# Discussion on Electron Temperature of Gas-Discharge Plasma with Non-Maxwellian Electron Energy Distribution Function Based on Entropy and Statistical Physics

**DOI:** 10.3390/e25020276

**Published:** 2023-02-02

**Authors:** Hiroshi Akatsuka, Yoshinori Tanaka

**Affiliations:** 1Laboratory for Zero-Carbon Energy, Institute of Innovative Research, Tokyo Institute of Technology, 2-12-1-N1-10, O-Okayama, Meguro-ku, Tokyo 152-8550, Japan; 2Department of Energy Sciences, Tokyo Institute of Technology, 2-12-1-N1-10, O-Okayama, Meguro-ku, Tokyo 152-8550, Japan

**Keywords:** non-equilibrium plasma, electron energy distribution function (EEDF), electron temperature, mean energy, Gibbs entropy, statistical physics

## Abstract

Electron temperature is reconsidered for weakly-ionized oxygen and nitrogen plasmas with its discharge pressure of a few hundred Pa, with its electron density of the order of $10^{17}\;m^{-3}$ and in a state of non-equilibrium, based on thermodynamics and statistical physics. The relationship between entropy and electron mean energy is focused on based on the electron energy distribution function (EEDF) calculated with the integro-differential Boltzmann equation for a given reduced electric field E/N. When the Boltzmann equation is solved, chemical kinetic equations are also simultaneously solved to determine essential excited species for the oxygen plasma, while vibrationally excited populations are solved for the nitrogen plasma, since the EEDF should be self-consistently found with the densities of collision counterparts of electrons. Next, the electron mean energy *U* and entropy *S* are calculated with the self-consistent EEDF obtained, where the entropy is calculated with the Gibbs’s formula. Then, the “statistical” electron temperature $T_{e}^{\mathrm{st}}$ is calculated as $T_{e}^{\mathrm{st}}={\left[\partial S/\partial U\right]}^{-1}$. The difference between $T_{e}^{\mathrm{st}}$ and the electron kinetic temperature $T_{e}^{\mathrm{kin}}$ is discussed, which is defined as [2/(3k)] times of the mean electron energy U=〈ϵ〉, as well as the temperature given as a slope of the EEDF for each value of E/N from the viewpoint of statistical physics as well as of elementary processes in the oxygen or nitrogen plasma.

## 1. Introduction

For high-temperature and high-density plasmas such as atmospheric-arc discharge plasmas, assumption of thermodynamic equilibrium holds almost exactly, and consequently, fluid-dynamic approximation can be applied with sufficient accuracy. Therefore, basic research on the collective phenomena, including relaxation processes, has produced various results [1]. Similarly, for industrial applications of low-temperature, approximately ∼1–4 eV, low-density ∼$10^{9}$–$10^{12}\;{\mathrm{cm}}^{-3}$, and non-equilibrium plasma systems, it is important to understand the fundamentals of such collective phenomena. However, in the low-temperature and low-density plasmas, almost all the constituents, such as electrons, positive and negative ions, reactive radicals, photons, etc., are non-uniformly distributed in the space with various time constants, and in short, they are in a state of non-equilibrium [2]. Since the collision frequency of each particle is small there, the electrons accelerated by the electric field do not sufficiently reach the thermodynamic equilibrium with atoms or ions. Therefore, while the electron temperature of plasmas generated in general experimental laboratories is tens of thousands of degrees Celsius, the temperature of ions and neutral particles remains approximately at room temperature for industrial applications, e.g., in electronic engineering such as semiconductor etching, thin-film deposition [3,4], or material processing such as plasma surface modification [5]. As a result, the electron energy distribution usually deviates from the Maxwell-Boltzmann distribution law [6,7]. That is, the processing plasma of lab-scale is generally in a state of non-equilibrium. Consequently, the simple treatment such as thermal equilibrium plasmas in which the energies of all the constituent particles can be described by a single temperature *T* becomes impossible.

However, even in a plasma system with such a high-degree of non-equilibrium, there are various methods to evaluate the thermodynamic properties of the plasmas comprehensively, one of which is to understand the electron energy distribution function (EEDF), and those of various excited species [3]. By evaluating the thermodynamic quantities of the constituents, it is considered that the unique structure and a certain order can be confirmed to describe the collective phenomena. Among them, the electron populations play the most important role in determining the characteristics of low-temperature, non-equilibrium plasma systems, and in particular, high-energy electron groups often play the most essential role. Moreover, the EEDF of the plasma is calculated accurately by reflecting the information of the elementary processes that are considered to occur in the plasma [7,8]. Therefore, it is considered that the analysis of the EEDF will greatly improve the understanding of the collective characteristics as well as the elementary chemical processes in plasmas. For example, if the EEDF is known, it can determine not only the macroscopic parameters, such as electron density and electron average energy, but also the various rate coefficients of various excitations, dissociations of neutral molecules and even ionization, which can provide essential guidelines for process application of the non-equilibrium plasmas [9].

The concepts of “temperature” in equilibrium and that in non-equilibrium are different, and especially in the latter case, care must be taken in the use and definition of the expression “temperature”. It goes without saying that the electron temperature and that of heavy species are different, and this temperature difference causes non-equilibrium to protect the process target from thermal damage. In addition, due to the deviation of the EEDF from the Maxwellian, it is necessary to pay attention to the fact that integral calculation using the cross-section data is required in the calculation of various rate coefficients to evaluate the densities of reactive species in the plasma [3,6,7,8,9]. These facts show that the consideration of “electron temperature” defined by the electron population is important in terms of basic physics.

It has been well known that the EEDF *F* becomes far from a Maxwellian one in weakly-ionized plasma for various processing applications, partly because the elastic and inelastic collisions with neutral species become predominant to the Coulomb collision. This kind of evaluation of the EEDF has been frequently reported based on the interest in the gas discharge physics [10,11,12]. For example, the high-energy component of EEDF is depleted for argon plasma with its discharge pressure of approximately ∼100 Pa due to frequent excitation of neutral argon atoms, which was experimentally observed with Langmuir probe measurement for CCP discharge and for microwave discharge [10,13]. On the other hand, numerically, the EEDF can be calculated as a solution to the integro-differential Boltzmann equation (hereafter, referred to as the Boltzmann equation, which will be later specified with its definition as Equations (Equation 23)–(Equation 26)) with two-term approximation for the given applied reduced electric field E/N in several gas discharge species, for which even free software can be obtained through web pages, e.g., BOLSIG+ [14].

Meanwhile, Alvarez et al., discussed the definition of rigorous electron temperature even for the plasma with non-Maxwellian EEDF in a state of non-equilibrium [15]. Alvarez et al., applied the software BOLSIG+ (Ver. 06/2013) to several gas discharge plasmas and discussed the relationship between the electron mean energy U≡〈ϵ〉 and the entropy *S*. They claimed that they found a common relationship $T_{e}\equiv {\left[\partial S/\partial U\right]}^{-1}=\left[2/\left(3k\right)\right]\langle \epsilon \rangle$, for any kinds of discharge species with non-Maxwellian EEDF, where *k* is the Boltzmann constant, which should be reconfirmed. It should be further remarked that Alvarez et al., did not consider the variation in the constituents of discharge species as counterparts of the electron collision, which becomes essential for the evaluation of super-elastic collisions. Namely, the rate equations of the essential excited species in the discharge should be simultaneously solved with the Boltzmann equation, which had been reported for many atmospheric discharge species [7,16,17,18]. If a rigorous EEDF is necessary as a solution to the Boltzmann equation, the chemical kinetics of discharge species must be also solved for the precise evaluation of the collision terms.

For example, if the electron kinetics are treated in weakly ionized O${}_{2}$ or N${}_{2}$ gas discharge plasmas to obtain the EEDF, the dissociation degree of O${}_{2}$ molecules or the vibrational distribution of N${}_{2}$ molecule must be considered because of the variation in the inelastic collisions of electrons. For instance in the oxygen plasma, if the value of E/N is increased as one of the input parameters, the dissociation degree becomes larger, and consequently, the electron collisions with O atoms should become essential in comparison with those with O${}_{2}$ molecules, which results in the variation in the EEDF [19,20,21,22,23]. If we treat nitrogen plasma, we must notice that the vibrational distribution function (VDF) of N${}_{2}$ X state has strong coupling with the EEDF of the plasma, which should be corrected in accordance with the VDF [7,17,18,22,24,25,26]. However, Alvarez et al., did not consider these kinds of variation in the collision partners of electrons as back ground gas species [15]. Based on the backgrounds described so far, the objective of the present study is to reconsider the “electron temperature” of the non-equilibrium plasma with non-Maxwellian EEDF from the viewpoint of statistical physics by applying the concept of entropy. As Alvarez et al., studied from the viewpoint of statistical physics, the temperature is given as a reciprocal value of the partial derivative of the entropy with respect to the internal energy. On the other hand, if the distribution function is known, the entropy of the system can be calculated, without concepts on equilibrium physics such as “free energy”. Thus, examination on the relation between the entropy and the energy through the distribution function can give another concept of the electron temperature of the non-equilibrium plasma, which is precisely the objective of the present study.

## 2. Theoretical Backgrounds

As case studies, an oxygen plasma and a nitrogen plasma are chosen in this study. For the oxygen discharge plasma, the essential chemical kinetics of oxygen atoms and molecules are simultaneously solved with the Boltzmann equation including dissociation degree of oxygen molecule [23], while the vibrational distribution function of the N${}_{2}$ molecule is solved simultaneously for the nitrogen plasma to include the effect of the superelastic collision of electrons with the vibrational levels [26].

### 2.1. Thermodynamics and Statistical Physics of Electrons

First, from the viewpoint of thermodynamics and statistical mechanics, the definition of “temperature” should be reconfirmed. The current definition of temperature *T* in physics is the differentiation of the kinetic energy *U* of a species in equilibrium by a statistical value called entropy *S* [27]. Since the equilibrium states in the true sense of the word are rare in nature, the temperature is defined for convenience even for the non-equilibrium state. At present, the definition of temperature in the non-equilibrium state is in the process of being defined because it may not be possible to define it in the original sense. Therefore, the definition of temperature is confirmed here based on the conventional theory.

Then, the first law of thermodynamics is confirmed. The kinetic energy of electron gas *U* is given by the so-called electron mean energy. Then, we denote the electron entropy *S*, its pressure *p*, its volume *V*, its chemical potential μ, and its total number *n*. The first law of thermodynamics is described as follows for a closed system [27,28]:(1)$$dS=\frac{1}{T}\left(dU+pdV-\mu dn\right).$$

In this research, only the free electron group is focused on in the plasma, ignoring the energy of ions and neutral particles. The excitation energy of the generated atomic molecules is also neglected for the time being. Since only the laboratory non-equilibrium plasmas are being targeted such as those obtained by small current discharge, the temperature of ions and neutral particles is much lower than that of electrons. Hence, the pressure of these heavy particles can be ignored. Since the number of electrons in the space is assumed to be constant and the volume change is not considered, the electron density can also be considered to be constant. Of course, from Equation (Equation 1), the following equations become obvious: (2)∂S∂UV,n=+1T,(3)∂S∂VU,n=+pT,(4)$$\begin{array}{ccc} {\left(\frac{\partial S}{\partial n}\right)}_{V,U} & = & -\frac{\mu }{T}. \end{array}$$
If the electron population performs electrical or mechanical work, it must be incorporated into energy conservation. However, for the sake of simplification of the discussion that follows, the volume change is neglected, that is, it is assumed that dV=0. Electrons are to move in a given DC electric field with its reduced electric field E/N. Neither the polarization effect of the charged species nor the plasma oscillation is considered. It is also assumed that the electron kinetic energy acquired by the change of space potential shall be exactly lost by collisions with the neutral particle as well as by any kind of radiation. That is, the change in spatial potential need not be described explicitly by the energy conservation relation. Furthermore, it is assumed that the electron density $N_{e}$ is also constant, then it leads to $dN_{e}=0$ when dV=0, that is, the total number of electrons *n* does not change, i.e., dn=0, and consequently, the chemical potential need not be considered throughout this study. To fulfill this condition, the discharge condition applied in this study is controlled so that the electron density may be kept constant.

As a result, in this study, the following equation is assumed without its validation but basically from analogy of the thermodynamics of thermal systems:(5)$$dS=\frac{1}{T}dU.$$
That is, in order to obtain the temperature *T*, the entropy *S* and the kinetic energy *U* should be obtained under the given conditions, and the relationship between these *S* and *U* should be investigated in detail under the above assumption, where the strategy is the same as Alvarez et al. [15].

In the meantime, the internal energy and entropy of the free electron group are calculated as follows. By applying the knowledge of statistical physics, it is possible to find a relation with conventional plasma physics. That is, in the field of plasma or gas discharge, various discussions are being held on the EEDF F(ϵ) or the electron energy probability function (EEPF) $f\left(\epsilon \right)\equiv F/\sqrt{\epsilon }$ by solving the Boltzmann equation by two-term approximation as a function of the reduced electric field E/N, where ϵ is the electron energy, *E* is the electric field and *N* is the number density of the neutral particles [3,6,7,8,9].

Here, the definition of the EEDF should be confirmed. First, the electron velocity distribution function g(v) is defined in this study as follows:(6)$$\begin{array}{cc} g_{3}\left(v\right)d^{3}v & =g_{3}\left(v_{x},v_{y},v_{z}\right)dv_{x}dv_{y}dv_{z}=g\left(v\right)\times 4\pi v^{2}dv\\ & =\;\mathrm{number}\mathrm{of}\mathrm{electrons}\mathrm{inside}a\mathrm{three}-\mathrm{dimensional}\mathrm{velocity}\mathrm{space}\\ & \;\;\mathrm{with}\mathrm{its}\mathrm{volume}\;dv_{x}dv_{y}dv_{z}\;\mathrm{at}\;v\;\mathrm{divided}\mathrm{by}\mathrm{total}\mathrm{number}\mathrm{of}\mathrm{electrons}. \end{array}$$
Namely, the distribution function $g_{3}\left(v_{x},v_{y},v_{z}\right)$ is uniformly distributed. However, the scalar-velocity distribution function g(v) is normalized as
(7)$$\int_{0}^{\infty }g\left(v\right)\times 4\pi v^{2}dv=1,$$
and in this respect, the velocity interval (v,v+dv) is statistically weighted by $4\pi v^{2}$.

Then, in the present study, the definition of the corresponding EEDF, F(ϵ), is defined as follows [3,29]:(8)$$4\pi v^{2}g\left(v\right)dv\equiv F\left(\epsilon \right)d\epsilon ,$$
and for electron with its mass $m_{e}$, as a matter of course,
(9)$$\epsilon =\frac{1}{2}m_{e}v^{2},$$
and from Equations (Equation 7) and (Equation 8), EEDF F(ϵ) is normalized as
(10)$$\int_{0}^{\infty }F\left(\epsilon \right)d\epsilon =1.$$
For electrons with Maxwellian EEDF and temperature $T_{e}$, it is well known that g(v) and F(ϵ) become $g_{M}\left(v\right)$ and $F_{M}\left(\epsilon \right)$ as follows, respectively [29],
(11)$$\begin{array}{ccc} g_{M}\left(v\right)\times 4\pi v^{2}dv & = & {\left(\frac{m_{e}}{2\pi kT_{e}}\right)}^{3/2}\exp\left(-\frac{m_{e}v^{2}}{2kT_{e}}\right)\times 4\pi v^{2}dv, \end{array}$$
(12)$$\begin{array}{ccc} F_{M}\left(\epsilon \right)d\epsilon & = & \frac{2}{\sqrt{\pi }}{\left(\frac{1}{kT_{e}}\right)}^{3/2}\exp\left(-\frac{\epsilon }{kT_{e}}\right)\sqrt{\epsilon }\cdot d\epsilon , \end{array}$$
which are valid only for the equilibrium condition. As Equation (Equation 12) shows, the equilibrium EEDF has the weighting factor $\sqrt{\epsilon }$ in its form in addition to the Boltzmann factor $\exp[-\epsilon /\left(kT_{e}\right)]$. Therefore, to draw the EEDF on a semilogarithmic plot for convenience, the electron energy probabilistic function (EEPF) f(ϵ) has been frequently defined as follows [30,31,32,33]:(13)$$f\left(\epsilon \right)\equiv \frac{F(\epsilon )}{\sqrt{\epsilon }}.$$
Then, the Boltzmann plot, i.e., the semi-logarithmic plot of the electron population against the electron energy ϵ becomes linear for the Maxwellian EEDF, Equation (Equation 12). On the other hand, if the Boltzmann plot of the EEPF f(ϵ) is not linear, it indicates that the electrons are not in the state of equilibrium. It should be also remarked that the EEPF is related to the velocity distribution function g(v) as the following:(14)$$f\left(\epsilon \right)=2\pi {\left(\frac{2}{m_{e}}\right)}^{3/2}g\left(v\right),$$
and consequently, the EEPF *f* is found to be a constant multiple of *g*.

If *F* is known, the internal energy *U* can be considered as the average energy of the electron group, and it can be calculated as follows:(15)$$U\equiv \langle \epsilon \rangle =\int_{0}^{\infty }\epsilon F\left(\epsilon \right)d\epsilon .$$

On the other hand, the electron entropy can be calculated with integral calculation by the Gibbs’s entropy formula established to a continuous variable rather than a discrete type [27]:(16)$$S=-k\int_{0}^{\infty }F\left(\epsilon \right)\ln\left[f(\epsilon )\right]d\epsilon .$$
Since the statistical weight of the energy interval (ϵ,ϵ+dϵ) is $\sqrt{\epsilon }$ as shown in Equations (Equation 12) and (Equation 13), the argument of the function ln in Equation (Equation 16) must be the EEPF f(ϵ), not the EEDF F(ϵ). The validity of Equation (Equation 16) will be confirmed in the final discussion of this paper.

When the EEDF is Maxwellian as defined in Equation (Equation 12), its internal energy $U_{M}$ and the Maxwellian entropy $S_{M}$ are analytically calculated as follows, respectively: (17)$$\begin{array}{lll} U_{M} & = & \int_{0}^{\infty }\epsilon F_{M}\left(\epsilon \right)d\epsilon =\frac{3}{2}kT_{e},\;\;\;\;\;\;\;\;\;\; \end{array}$$(18)$$\begin{array}{lll} S_{M} & = & -k\int_{0}^{\infty }F_{M}\left(\epsilon \right)\ln\left[f_{M}\left(\epsilon \right)\right]d\epsilon =\frac{3k}{2}\left[1+\ln\left(\frac{4\pi U_{M}}{3m_{e}}\right)\right]. \end{array}$$
Then, Equations (Equation 17) and (Equation 18) lead to
(19)$$\frac{\partial S_{M}}{\partial U_{M}}=\frac{3k}{2U_{M}}=\frac{1}{T_{e}}.$$
Hence, for the Maxwellian EEDF, Equation (Equation 12), the validity of Equation (Equation 2) is confirmed, which is also consistent with the definition of the entropy by Equation (Equation 16).

That is, in this study, first, the gas type was fixed to oxygen or nitrogen, and then, after a reduced electric field E/N was given, EEDF F(ϵ) was obtained by the Boltzmann analysis. Subsequently, the energy *U* and the entropy *S* are obtained by Equations (Equation 15) and (Equation 16), respectively. In other words, the relational expression between *S* and *U* can be obtained by changing the reduced electric field E/N. Then, the relationship with the “electron temperature” obtained from Equation (Equation 2), i.e., ${\left[\partial S/\partial U\right]}^{-1}=T_{e}$ is discussed in this study [15].

As a similar previous study, Alvarez et al., has already developed the above discussion on the pure discharge plasmas of Ar, He, N${}_{2}$, O${}_{2}$, and H${}_{2}$ [15]. Using the free software BOLSIG+ [14], Alvarez et al., solved the corresponding Boltzmann equation and obtained their EEDF based on two-term approximation, and drew the following conclusions. That is, Alvarez et al., claimed for any of the foregoing discharge gas species that the relation
(20)$$S=S_{0}+\frac{3k}{2}\ln U$$
held, and consequently, the following relationship
(21)$$T_{e}={\left(\frac{\partial S}{\partial U}\right)}^{-1}=\frac{2U}{3k}\;i.e.,\;U=\frac{3}{2}kT_{e},$$
was found. Additionally they concluded that the above equation is universally valid even when the EEDF does not follow the Boltzmann distribution, Equation (Equation 12) [15].

However, their theory has some serious problems. The first problem is the shape of the EEDF they calculated. Their EEDF’s are far from the Maxwellian distribution for any discharge gas. It is incredible that Equations (Equation 20) and (Equation 21) can be obtained in such cases. The points that may require further reverification are as follows. There is a crucial physical process that is omitted in their model to calculate the EEDF of the actual discharge plasma. Specifically, in the plasma of molecular gas species such as oxygen and nitrogen, the state of the target gas should change as a result of the dissociation of molecules and the existence of vibrationally excited states, which have not been incorporated in their calculation. Changes in the excited state distribution of these targets are not taken into account. More specifically, in the case of O${}_{2}$, dissociation should proceed as the E/N increases. Depending on the degree of dissociation, the rate of O${}_{2}$ molecule excitation in the energy loss process should decrease, and instead collision with O atoms should be predominant. When dealing with N${}_{2}$, the VDF of N${}_{2}$ X state changes, and the amount of energy change in inelastic collision changes according to the VDF under each discharge condition. Then, the effectiveness of the superelastic collision should be reflected in EEDF in the nitrogen plasma. However, these changes in target species are not included, which leaves problems with the calculations of Alvarez et al. [15].

On the other hand, in the research field of low-temperature plasma, the mainstream is the method of establishing a global model for the density change of the excited species and solving it simultaneously with the Boltzmann Equation [7,16,17,18,19,34,35]. Rather, changes in the excited-state populations as collision counterparts such as oxygen and nitrogen are focused on, which are important for the formation of EEDF. Studies have been conducted on the kinetics of excitation kinetics. From such past accumulation, it is possible to obtain a self-consistent EEDF by coordinating with the change in excited-state density [21,22,23,24,25,26,36].

Therefore, in this study, the conventional model is applied to find both the number density of the major excited states in oxygen or nitrogen plasma and the EEDF as a function of the reduced electric field in a self-consistent manner. Therefore, the EEDF is calculated according to this model, and then the “electron temperature” obtained statistically and thermodynamically is calculated using Equations (Equation 2), (Equation 15) and (Equation 16). Then, the obtained electron temperature will be compared and discussed with the slope of the EEPF obtained for each E/N. Regarding the electron temperature of plasmas with non-Maxwellian EEDF, the main objective of this study is to deepen the mathematical-physics study on the validity of Equations (Equation 15) and (Equation 16), as well as the deeper mechanism for describing such non-equilibrium electron statistics.

### 2.2. Confirmation of the Boltzmann Equation to Be Solved

It should be emphasized that the EEPF f(ϵ) determined in Equation (Equation 13) has long been applied in the two-term approximated Boltzmann equation instead of the EEDF [30,31,37]. These references showed that the velocity distribution function g(v) under the external electric field along the *z*-axis is well described with the two-term approximation, that is, the summation of the isotopic component of *g* as $g_{0}$ and the first-order anisotropic component $g_{1}$:(22)$$g\left(v\right)=g_{0}\left(v\right)+\frac{v_{z}}{v}g_{1}\left(v\right)$$
On the other hand, the velocity distribution function g(v) is given as a solution to the Boltzmann equation as follows:(23)$$\frac{\partial g}{\partial t}+v\cdot \nabla_{r}\;g-\frac{eE}{m_{e}}\cdot \nabla_{v}\;g={\left.\frac{\delta g}{\delta t}\right|}_{\mathrm{coll}.},$$
where *e* is the elementary charge, E is the electric field, and ${\delta g/\delta t|}_{\mathrm{coll}}.$ is the collision term. In the present analysis, since the electrons do not depend on spatial coordinates, $\nabla_{r}\equiv 0$. It can be also assumed that $m_{e}\ll m_{i},M$ where $m_{i}$ and *M* are the mass of ions and that of neutral particles, respectively, and that $T_{e}\gg T_{g}$ where $T_{g}$ is the gas temperature, which are common to low-temperature plasmas. Thus, by substitution of Equation (Equation 22) into Equation (Equation 23), multiplying the respective Legendre polynomials (1 and cosθ) and integrating over cosθ, the following two differential equations are obtained for the isotopic component of the EEPF $f_{0}$ and the directional component $f_{1}$ instead of *g* by applying Equation (Equation 14) [14,30,31,37,38]:(24)∂f0∂t=132eme1/2Eϵ1/2∂∂ϵ(ϵf0)+1ϵ1/2∂∂ϵϵ3/22meMνel(ϵ)f1=+∑jϵ+ϵsi−jϵνjin(ϵ+ϵsi−j)f0(ϵ+ϵsi−j)−νjin(ϵ)f0(ϵ),
(25)$$\frac{\partial f_{1}}{\partial t}={\left(\frac{2e\epsilon }{m_{e}}\right)}^{1/2}E\frac{\partial }{\partial \epsilon }\left(f_{0}\right)-\left[\nu^{\mathrm{el}}\left(\epsilon \right)+\sum_{j}\nu_{j}^{\mathrm{in}}\left(\epsilon \right)\right]f_{1}\left(\epsilon \right),$$
where E=|E|, absolute value of the electric field, $\nu^{\mathrm{el}}\left(\epsilon \right)$ is the elastic collision frequency of electron of energy ϵ with neutral particles, $\nu_{j}^{\mathrm{in}}\left(\epsilon \right)$ is that of the *j*-th type electron inelastic collision, and $\epsilon_{\mathrm{si}-j}$ is the energy loss at the corresponding inelastic collision. In the present study, the electrons in the plasma are assumed to be in the steady state, and consequently, ∂/(∂t)≡0. Then, $f_{1}$ is eliminated from Equations (Equation 24) and (Equation 25), and eventually, the second-order ordinary differential equation of the isotopic component of EEPF, $f_{0}$, with respect to the electron energy ϵ is derived as
(26)$$\begin{array}{c} -\frac{d}{d\epsilon }\left[\frac{1}{3}{\left(\frac{E}{N}\right)}^{2}\frac{\epsilon }{\sigma_{c}\left(\epsilon \right)+\sum_{j}\sigma_{\mathrm{si}-j}\left(\epsilon \right)}\frac{df_{0}}{d\epsilon }+\frac{2m_{e}}{M}\sigma_{c}\left(\epsilon \right)\epsilon^{2}\left(f_{0}+\frac{kT_{g}}{e}\frac{df_{0}}{d\epsilon }\right)\right]\;\\ \;+\sum_{j}\left\{\epsilon \sigma_{\mathrm{si}-j}\left(\epsilon \right)f_{0}\left(\epsilon \right)-\left(\epsilon +\epsilon_{\mathrm{si}-j}\right)\sigma_{\mathrm{si}-j}\left(\epsilon +\epsilon_{\mathrm{si}-j}\right)f_{0}\left(\epsilon +\epsilon_{\mathrm{si}-j}\right)\right\}=0, \end{array}$$
where *N* is the number density of the neutral species, $\sigma_{c}$ is the momentum-transfer cross section, $\sigma_{\mathrm{si}-j}$ is the cross section of the *j*-th type inelastic collision, and $T_{g}$ is the gas temperature. Of course, E/N is the reduced electric field. The first term of the first line of Equation (Equation 26) indicates the energy gain of electrons accelerated by the external electric field, while the second term corresponds to the energy loss by the elastic collision processes with neutral particles. Meanwhile, the second line shows the effect of inelastic collisions. The first term denotes the outgoing flow from the energy interval (ϵ,ϵ+dϵ) whereas the second term corresponds to the incoming kinetics to this interval by superelastic collisions. Hereafter, as there will be no risk of misunderstandings, the isotropic component of the EEPF, $f_{0}$, will be written simply as *f* for the sake of simplification [24,30,31,37,39].

### 2.3. Calculation of EEPF of Oxygen Plasma—Self-Consistent Simultaneous Solution with Rate Equations of Major Excited Species

The method is confirmed here to solve the EEDF of oxygen plasma simultaneously with the density of essential excited species in the oxygen plasma [19,20,21]. As the E/N increases and the average electron energy increases accordingly, the dissociation of oxygen molecules into atoms progresses. As a result, the dominant collision processes related to electron energy loss changes from the molecular kinetics to the atomic excitation kinetics, which is incorporated into the computational model.

Here, a low-pressure steady-state oxygen plasma with a discharge pressure of about 1 Torr is chosen to be an example of the discussion and treated by a global model [23]. In the case of such oxygen plasma, the following eight states are individually treated as the main states at the steady state: $O_{2}\;\left(X{\;}^{3}\Sigma_{g}^{-}\right),O_{2}\;\left(a{\;}^{1}\Delta_{g}\right),O_{2}\;\left(b{\;}^{1}\Sigma_{g}^{+}\right),O^{-},O_{3},O_{2}^{+},O\;{(}^{3}P)$, and $O\;{(}^{1}D)$. At this time, the input values of the numerical simulation to determine the densities of the above excited species are the discharge pressure *P* for determining the total particle density, the gas temperature $T_{g}$ for determining the particle density and the atomic/molecular reaction rate coefficient, the electron density $N_{e}$, the inner radius of the discharge tube *R* that becomes necessary for diffusion loss rates of some species, and the reduced electric field E/N to determine the EEDF.

A set of rate equations based on a global model are established for changes in the number density of each state. That is, they are rate equations described by the generation or loss due to the electron collision, the atomic-molecule collision reaction, or the diffusion loss for each state. In short, the following equations are obtained for each state.
(27)$$\frac{d[A]}{dt}=-\nu_{W}\left[A\right]+G,$$
where, [A] is the number density of the atomic or molecular species A to be considered, $\nu_{W}$ is the loss rate coefficient due to collision with the wall, and *G* is the source term indicating the generation or loss of atomic or molecular species A due to electron collision or atomic/molecular collision. $\nu_{W}$ is determined by the boundary condition at the discharge tube wall, that is, the deactivation probability γ at the time of wall collision. Assuming that the average thermal velocity of the particles is *c* and the diffusion coefficient of the particles is *D*, $\nu_{W}$ is calculated as follows: (28)$$\begin{array}{ccc} \nu_{W} & = & \frac{\gamma c}{2R}\;\mathrm{for}\;\gamma \ll 1, \end{array}$$(29)$$\begin{array}{ccc} \nu_{W} & = & {\left(\frac{2.405}{R}\right)}^{2}D\;\mathrm{for}\;\gamma ∼1. \end{array}$$

Next, the electron collision rate coefficient $k_{e}$ for the generation and loss is given by the integral as shown in the following equation,
(30)$$k_{e}=\int_{\epsilon_{\mathrm{th}}}^{\infty }\sigma \left(\epsilon \right)F\left(\epsilon \right)vd\epsilon =\int_{\epsilon_{\mathrm{th}}}^{\infty }\sigma \left(\epsilon \right)\left[\sqrt{\epsilon }\cdot f\left(\epsilon \right)\right]\cdot \sqrt{\frac{2\epsilon }{m_{e}}}d\epsilon =\sqrt{\frac{2}{m_{e}}}\int_{\epsilon_{\mathrm{th}}}^{\infty }\sigma \left(\epsilon \right)\epsilon f\left(\epsilon \right)d\epsilon ,$$
where σ(ϵ) is the cross section of the corresponding electron collision, $\epsilon_{\mathrm{th}}$ is the threshold energy of the reaction, and f(ϵ) is the EEPF defined as Equation (Equation 13). For the oxygen plasma, the two-term approximated Boltzmann Equation (Equation 26), should be modified as follows, because of the variation in the collision partner due to dissociation of oxygen molecules into oxygen atoms with the increase in the reduced electric field, and is solved by numerical calculation, the procedure of which was already specified in [23]: (31)$$\begin{array}{c} -\frac{d}{d\epsilon }\left[\frac{1}{3}{\left(\frac{E}{N}\right)}^{2}\frac{\epsilon }{\sum_{s}\delta_{s}\left\{\sigma_{c}^{s}\left(\epsilon \right)+\sum_{j}\sigma_{\mathrm{si}-j}^{s}\left(\epsilon \right)\right\}}\frac{df(\epsilon )}{d\epsilon }\right.\;\\ \;\;\left.+\epsilon^{2}\left\{\sum_{s}\delta_{s}\frac{2m_{e}}{M_{s}}\sigma_{c}^{s}\left(\epsilon \right)\right\}\left\{f\left(\epsilon \right)+\frac{kT_{g}}{e}\frac{df(\epsilon )}{d\epsilon }\right\}\right]\;\\ \;+\sum_{j,s}\delta_{s}\left\{\epsilon \sigma_{\mathrm{si}-j}^{s}\left(\epsilon \right)f\left(\epsilon \right)-\left(\epsilon +\epsilon_{\mathrm{si}-j}^{s}\right)\sigma_{\mathrm{si}-j}^{s}\left(\epsilon +\epsilon_{\mathrm{si}-j}^{s}\right)f\left(\epsilon +\epsilon_{\mathrm{si}-j}^{s}\right)\right\}=0, \end{array}$$
where $\sigma_{c}^{s}$ is the momentum-transfer cross section of the species *s* (s=1 for O atom and s=2 for O${}_{2}$ molecule), $M_{s}$ is the mass of the *s*-th target molecule of electron collision, $\delta_{s}$ is the number-density fraction of the *s*-th species, $\sigma_{\mathrm{si}-j}^{s}$ is the cross section of *j*-th type inelastic collision of the *s*-th heavy soecies, $\epsilon_{\mathrm{si}-j}^{s}$ is the energy loss at that inelastic collision, and $T_{g}$ is the gas temperature.

In this study, 15 reactions listed in Table 1 were considered when solving Equation (Equation 27) for the population or depopulation kinetics of chemical species as electron collision reactions, while another 15 types of reactions shown in Table 2 were considered as the atomic-molecular collision reaction of Equation (Equation 27).

Next, as the inelastic collisions of the collision term of the Boltzmann Equation (Equation 31) (the third line of Equation (Equation 31), the reactions shown in Table 3 were considered. In the Boltzmann analysis of oxygen plasma, the so-called superelastic collision, the last term of the left hand side of Equation (Equation 31), $-\left(\epsilon +\epsilon_{\mathrm{si}-j}^{s}\right)\sigma_{\mathrm{si}-j}^{s}\left(\epsilon +\epsilon_{\mathrm{si}-j}^{s}\right)f\left(\epsilon +\epsilon_{\mathrm{si}-j}^{s}\right)$, was neglected, because of the low absolute amount of the counterpart of superelastic collision. This effect has been found to be small enough. As will be described later, in the Boltzmann analysis of nitrogen plasma, superelastic collision with vibrationally excited state of N${}_{2}$ X state is extremely essential and cannot be ignored, which is different from the oxygen plasma. However, for the oxygen plasma, the degree of dissociation of O${}_{2}$ molecule changes as the calculation progresses. That is, not only the ground state $O_{2}\;\left(X{\;}^{3}\Sigma_{g}^{-}\right)$ of oxygen molecule but also the ground state $O\;{(}^{3}P)$ of oxygen atom exists as the target of electron inelastic collision. Therefore, it is necessary to quantitatively reflect the degree of dissociation of oxygen molecules when performing the calculation to solve Equation (Equation 31) by giving each value of E/N. Hence, the reactions shown in Table 3 should be included. The numerical solution of Equation (Equation 31) was obtained with 400 points as one interval of Δϵ in 0.1 eV increments for the energy range up to 0.1–40 eV.

Considering the collisions in Table 1, Table 2 and Table 3, the rate equations for the excited-state number densities Equation (Equation 27) and the Boltzmann Equation (Equation 31) are solved repeatedly until they become self-consistent with respect to the excited-state density and EEPF f(ϵ). Normally, the system becomes steady and self-consistent by repeated calculation several times, and the number density of excited states and the kinetics of each reaction can be obtained. Further details to calculate the densities of excited species in the oxygen plasma are specified in Konno et al. [23].

### 2.4. Calculation of EEPF of Nitrogen Plasma—Self-Consistent Simultaneous Solution with the Vibrational Distribution Function of N${}_{2}$
Electronically Ground State

For the nitrogen plasma, the basic equation for the EEPF is still Equation (Equation 31). However, as the essential difference of the N${}_{2}$ plasma from the O${}_{2}$ plasma, the superelastic collision of electrons with vibrationally excited N${}_{2}$ becomes quite essential in Equation (Equation 31), that is, the last term of the RHS. To calculate the effect of the superelastic collision with N${}_{2}$ vibrational level, the number density of the vibrational level must be known, i.e., the vibrational distribution function of N${}_{2}$ of the electronically ground level, N${}_{2}\;X{\;}^{1}\Sigma_{g}^{+}$. In other words, the *s*-th partner of inelastic collision in Equation (Equation 31) should be considered as each vibrationally-excited N${}_{2}$ molecule with its vibrational quantum number *v*.

A large number of research papers have been already published on the VDF of N${}_{2}$ molecule in low-pressure discharge nitrogen plasma [16,17,18,21,22,25,26,34]. They showed that the following elementary collision processes are essential to consider the rate equation of the N${}_{2}$ vibrational levels. That is, (1) e-V process; electron collision excitation and de-excitation, (2) V-V transfer; collisional energy transfer between two N${}_{2}$ molecules resulting in vibrational single-quantum exchange, (3) V-T relaxation; collisional energy relaxation of N${}_{2}$ vibrational single-quantum energy to the translational kinetic energy of another N${}_{2}$ molecules as a collision counterpart, and (4) V-limit dissociation; the dissociation through the vibrational limit. When the discharge pressure is about 1 or several Torr, the wall relaxation process of vibrational levels is almost negligible in comparison to the foregoing V-V or V-T processes. Due to small dissociation degree of nitrogen molecule, which is much smaller than that of oxygen indeed, the total number density of nitrogen molecules is assumed to be constant, $N(N_{2})$. The dissociated nitrogen atoms, in turn, are assumed to associate into the *v*-th vibrational level of N${}_{2}$ immediately after the dissociation with the same probability $R_{v}$ = const [26].

Based on the above assumptions, the VDF of N${}_{2}\;X{\;}^{1}\Sigma_{g}^{+}$ state is calculated from the following rate equation as number density of the *v*-th vibrational level $N_{v}$:(32)$$\begin{array}{cc} \frac{dN_{v}}{dt}= & \;N_{e}\sum_{w=0,\neq v}^{M}N_{w}C_{w}^{v}-N_{e}N_{v}\sum_{w=0,\neq v}^{M}C_{v}^{w}\;\\ \;\; & +N_{v-1}\sum_{w=0,\neq v}^{M-1}N_{w+1}Q_{v-1,v}^{w+1,w}+N_{v+1}\sum_{w=0,\neq v}^{M-1}N_{w}Q_{v+1,v}^{w,w+1}\\ \;\; & -N_{v}\left(\sum_{w=0,\neq v}^{M-1}N_{w+1}Q_{v,v+1}^{w+1,w}+\sum_{w=0,\neq v}^{M-1}N_{w}Q_{v,v-1}^{w,w+1}\right)\\ \;\; & +N\left(N_{2}\right)\left(N_{v-1}P_{v-1,v}+N_{v+1}P_{v+1,v}\right)\\ \;\; & -N\left(N_{2}\right)N_{v}\left(P_{v,v-1}+P_{v,v+1}\right)+R_{v}=0, \end{array}$$
with
(33)$$N\left(N_{2}\right)=\sum_{w=0}^{M}N_{w}=\mathrm{Const}.,$$
where $N_{e}$ is the electron density, $C_{v}^{w}$ is the electron collision excitation or deexcitation from *v* to *w* vibrational level, $Q_{v_{1},v_{2}}^{w_{1},w_{2}}$ is the rate coefficient of V-V transfer for the reaction
(34)$$N_{2}\left(v_{1}\right)+N_{2}\left(w_{1}\right)\to N_{2}\left(v_{2}\right)+N_{2}\left(w_{2}\right),$$
$P_{v_{1},v_{2}}$ is that of V-T relaxation for the reaction
(35)$$N_{2}\left(v_{1}\right)+N_{2}\to N_{2}\left(v_{2}\right)+N_{2},$$
and $R_{v}$ is that of atomic nitrogen recombination into the *v*-th vibrational level. In the present treatment of vibrational kinetics, only a single-vibrational quantum transfer is taken into account. That is, for Equations (Equation 34) and (Equation 35), reactions only with $|v_{1}-v_{2}|=1$ and $|w_{1}-w_{2}|=1$ are considered, as with always assumed in this kind of nitrogen vibrational kinetics [7]. The upper limit of the summation in Equation (Equation 32) is set to be M=46 as the dissociation limit. Table 4 summarizes the reactions relevant to the vibrational kinetics together with the list of references of the corresponding rate coefficients.

The master vibrational Equation (Equation 32) is solved simultaneously with the Boltzmann Equation (Equation 31) for the nitrogen discharge plasma, until the steady state is obtained together with the self-consistent EEDF-VDF. However, when the Boltzmann equation is solved for the nitrogen plasma, the variable *s* in Equation (Equation 31) is interpreted as the vibrational level *v*. Additionally for electron elastic collision with N${}_{2}$ molecule, its cross section $\sigma_{c}^{v}\left(\epsilon \right)$ is assumed to be the same function $\sigma_{c}\left(\epsilon \right)$ irrespective of the vibrational level *v*.

On the other hand, the effect of excitation kinetics of electronically excited states on the EEDF is considered to be rather minor to that of the N${}_{2}$ vibrational kinetics, because the number density of the electronically excited states of N${}_{2}$ molecule or that of atomic nitrogen is much smaller than that of vibrationally excited states of $N_{2}\;\left(X{\;}^{1}\Sigma_{g}^{+}\right)$. Furthermore, owing to the low dissociation degree of N${}_{2}$ molecule, in the calculation of EEDF of the nitrogen plasma, neither the number densities of excited species nor those of atomic species are not treated for the nitrogen plasma. Rather, the VDF should be essentially considered throughout the present calculation.

Meanwhile, as the inelastic collisions of the collision term of the Boltzmann Equation (Equation 31) of the nitrogen plasma, the reactions listed in Table 5 were considered. Just like the EEDF of the oxygen plasma, that of the nitrogen plasma is similarly calculated from Equations (Equation 31) and (Equation 32). The system also comes to the steady and self-consistent state with multiple iteration. Further details for the calculation of self-consistent EEDF and VDF of the nitrogen plasma with its discharge pressure of several Torr are specified in Sakamoto et al. [24].

## 3. Results and Discussion

### 3.1. Oxygen Plasma

Based on the principle described in Section 2.3, the EEPF of oxygen plasma can be obtained as a function of E/N as shown in Figure 1, where the gas temperature and the electron density are fixed to be $T_{g}=0.15$ eV and $N_{e}=2\times 10^{11}\;{\mathrm{cm}}^{-3}$, respectively, and the calculation is performed within the range of the reduced electric field 90≤E/N[Td]≤170, which fulfills realistic experimental conditions of oxygen plasma generated with common discharge devices. Of course, since the electron density is dependent on the initial gas pressure, the gas temperature is so adjusted to make the electron density becomes $N_{e}=2\times 10^{11}\;{\mathrm{cm}}^{-3}$ for each reduced electric field condition. As was described in Section 2.1, in the present calculation, the electron density should be kept constant, and the calculation is conducted to fulfill this requirement, $N_{e}=$ Const. The gas pressure was controlled to be 1 Torr by adjusting the gas temperature over the range $1000\leq T_{g}\;\left[K\right]\leq 2000$, which gave a realistic discharge condition in the lab-scale experiment.

As is clear from Figure 1, the EEPF of the oxygen plasma of this study is not Maxwellian, but is such that the high energy part is depleted more than Maxwellian EEDF [23]. That is, the high energy tail portion is colder than the bulk portion. The energy that separates the high-energy tail and the low-energy bulk part is found at approximately 7–8 eV, which is considered to be due to the large cross section of inelastic collision to generate atomic oxygen near this energy, which corresponds to reaction “35” in Table 3. In the electron collisional dissociation reactions, the reaction “12” in Table 1, whose threshold energy is ϵ=8.4 eV, has larger cross section than the generation of two ground-state oxygen atoms with lower threshold energy ϵ=5.7 eV [40]. It can also be understood from Figure 1 that the electron temperature defined as the slope of the EEPF hardly depends on the reduced electric field for the bulk portion of the EEPF, which is found in the lower electron energy region, ϵ≲8.4 eV. Hereafter, the boundary between the bulk and the tail is treated to be 8.4 eV.

Next, rather intuitively, it can be considered that the slope of EEPF shown in Figure 1 indicates the temperature at each electron energy ϵ. Therefore, this differential value is referred to here as the “local electron temperature” $T_{\mathrm{loc}}$, where the “local” means that this temperature is defined at each local electron energy ϵ:(36)$$T_{\mathrm{loc}}\left(\epsilon \right)\equiv -\frac{1}{k}{\left[\frac{d\ln f(\epsilon )}{d\epsilon }\right]}^{-1},$$
which is shown in Figure 2. This figure indicates that the bulk component of electrons has a local electron temperature of approximately $T_{\mathrm{loc}}$= 1.4–5.5 eV, where the elastic collision reaction is considered to be dominant among the collisional processes. The bulk component seems to have a constant local electron temperature in Figure 1, however, Figure 2 shows that it is not the case. It is found that the bulk component has a maximum at the electron energy ϵ=5 eV, and it is confirmed that the electrons are in the state of non-equilibrium in any case. On the other hand, it can be seen that the local electron temperature in the high-energy tail region does not change much. It is confirmed that the local electron temperature of hight-energy tail region raises with increasing reduced electric field, however, it does not change as much as the bulk component.

Meanwhile, Figure 3 shows the results of the dissociation degree of oxygen molecules in the oxygen plasma plotted against the reduced electric field, where it is shown that the relative density of the atomic species among the heavy species increases almost linearly with increasing the reduced electric field. This effect, that is, the oxygen molecular dissociation into atoms was not included in the Alvarez et al.’s similar previous study [15], and the most original improvement in the present study. The effect of the existence of atomic oxygen on the EEPF should be confirmed, which was shown in Figure 4 for the reduced electric field E/N=110 Td as an example. Obviously, the discrepancy in the electron energy region ϵ∼0–2 eV is remarkable in addition to that in the high-energy tail region ϵ≥25 eV. In the very low energy range, 0≤ϵ[eV]≲2, when the existence of atomic oxygen is neglected and all the heavy species are assumed to be oxygen molecules, the energy loss of electron due to inelastic collision of rotational as well as vibrational excitation of O${}_{2}$ molecule is much more exaggerated than the real situation. Since the behavior of the EEPF in the low-energy region has strong impact on the calculation of the entropy by Equation (Equation 16), the evaluation of the dissociation degree of oxygen is quite essential, which was confirmed by the present calculation.

Next, various temperatures are compared with each other. Particularly, the temperature calculated by Equation (Equation 21) must be discussed in this study, which means that the values of the entropy of electrons *S* and their internal energy *U* as the electron mean energy 〈ϵ〉, must be specified. Then, the entropy of the electrons *S* should be calculated by Equation (Equation 16), which is shown in Figure 5, whereas the internal energy U=〈ϵ〉 of the electrons can be calculated by Equation (Equation 15) as illustrated in Figure 6. Therefore, the relation between the values *S* and *U* is found as in Figure 7. And finally, the temperature defined from the viewpoint of statistical thermodynamics can be calculated with Equation (Equation 2). Hereafter, the electron temperature calculated by Equation (Equation 2) will be referred to as “statistical electron temperature”, $T_{e}^{\mathrm{st}}$, which will be shown in Figure 8 with other temperatures which will be specified later on.

In the meantime, Alvarez et al. [15] as well as other researchers referred the 2/(3k) times of the electron mean energy 〈ϵ〉 as the kinetic temperature [13,22], and in this paper also, this value is defined as the “kinetic electron temperature”, $T_{e}^{\mathrm{kin}}$, which is of course,
(37)$$T_{e}^{\mathrm{kin}}\equiv \frac{2}{3k}\langle \epsilon \rangle .$$
Moreover, in order to discuss the behavior of the electron temperatures a little more quantitatively, the bulk temperature $T_{e1}$ and the tail temperature $T_{e2}$ are defined as follows. $T_{e1}$ is the maximum value of $T_{\mathrm{loc}}$ for each reduced electric field in the electron-energy interval of 0≤ϵ≤8.4 eV, while $T_{e2}$ is defined as the mean value of $T_{\mathrm{loc}}$ in the electron energy range ϵ≥8.4 eV. Consequently, Figure 8 summarizes the temperatures $T_{e1},T_{e2},T_{e}^{\mathrm{kin}}$ and $T_{e}^{\mathrm{st}}$, plotted against the reduced electric field E/N. From this figure, it is found that the statistical electron temperature $T_{e}^{\mathrm{st}}$ defined by the thermodynamic relation and the kinetic temperature $T_{e}^{\mathrm{kin}}$ can be interpreted as the weighted average electron temperature of the bulk electron temperature $T_{e1}$ and the tail electron temperature $T_{e2}$. In any case, it can be seen that these four temperatures have different values.

In the oxygen plasma in this study, all of the four electron temperatures, $T_{e1},T_{e2},T_{e}^{\mathrm{kin}}$ and $T_{e}^{\mathrm{st}}$, were found to be in the range of 1–5 eV, which is a reasonable temperature in a low-temperature process plasma with its discharge pressure of approximately ∼1 Torr. In any case, it was found for oxygen plasma that
(38)$$T_{e}^{\mathrm{kin}}≃\left(1.02-1.05\right)\times T_{e}^{\mathrm{st}}.$$
Therefore, it should be concluded that the statistical temperature is considered to agree rather precisely with the kinetic temperature, at least for the case of oxygen plasma, although the agreement is not exact and $T_{e}^{\mathrm{kin}}$ is always larger than $T_{e}^{\mathrm{st}}$ by (2–5)%. It should be also noted that the relationship between the entropy *S* of electrons and the electron mean energy U=〈ϵ〉 also approximately agrees with the one predicted by Equation (Equation 20), which is shown in Figure 9. This one indicates that d(S/k)/d(lnU)≃ 1.55–1.61 ≩1.5. This fact shows that the work of Alvarez et. al., can predict the relation between the entropy and the electron mean energy with its accuracy less than 10%, nevertheless, the agreement is still not perfect, probably due to the negligence of the dissociation of the oxygen molecule as the collision counterpart.

### 3.2. Nitrogen Plasma

Just like Figure 1 for the oxygen plasma, the EEPF of the nitrogen plasma was calculated based on the method described in Section 2.4, which is shown in Figure 10, under the condition of constant electron density $N_{e}=4\times 10^{11}\;{\mathrm{cm}}^{-3}$ and the total discharge pressure 1 Torr [24]. To keep this discharge condition, as was adopted in the calculation of the oxygen plasma, the gas temperature of the nitrogen plasma was also adjusted within the range of $1000\leq T_{g}\;\left[K\right]\leq 2000$ to keep the electron density constant, which is considered to be realistic discharge condition.

The stepwise decrease in the energy range ϵ≃ 2–3 eV is a typical characteristics of the EEPF of the nitrogen plasma, which is attributed to the electron energy consumption due to the inelastic collision to generate vibrationally excited levels of N${}_{2}$ molecule, which corresponds to the reaction “e-V” described in Table 4 [42]. However, this stepwise energy change is rather moderated in this figure due to the superelastic collisions to enhance the electron energy from the vibrational energy of N${}_{2}$ molecule, which was incorporated in the present calculation by the simultaneous calculation of the VDF, some of which are illustrated in Figure 11. Although the dependence of the VDF on the reduced electric field is rather minor in the lower-vibrational energy region, the effect of the VDF on the EEPF is confirmed through these calculations. This is confirmed by the fact that the EEPF cannot be calculated precisely without the effect of electron inelastic collisions as vibrational excitation and deexcitation of N${}_{2}$ molecules. Figure 12 shows the comparison of the EEPFs with and without the simultaneous calculation of the vibrational kinetics of N${}_{2}$ molecules described as Equation (Equation 32), which indicates how essential the vibrational kinetics is in the formation of EEPF in the nitrogen plasma. Hereafter, all the EEPFs of the nitrogen plasma will be treated with superelastic collisions with N${}_{2}$ vibrational levels.

Thus, with the calculated EEPF shown in Figure 12, the local electron temperature $T_{\mathrm{loc}}$ can be calculated in the same way as for the oxygen plasma in Figure 2, which is shown in Figure 13. The dependence of $T_{\mathrm{loc}}$ of N${}_{2}$ plasma on the electron energy ϵ is completely different from that of O${}_{2}$ plasma. The local temperature of N${}_{2}$ plasma has a deep minimum at ϵ∼2.2 eV owing to the inelastic collision for N${}_{2}$ vibrational excitation, followed by a broad maximum at ϵ∼4 eV, for any value of the reduced electric field considered in this study. For the nitrogen plasma, the former minimum $T_{\mathrm{loc}}$ at ϵ∼2.2 eV is defined to be $T_{e1}$ as a representative minimum value, while the latter maximum value is $T_{e2}$ as a representative maximum value of the bulk-energy region. Concerning that of the tail region, $T_{\mathrm{loc}}\left(30\;\mathrm{eV}\right)$ will be referred in the later discussion. Meanwhile, in the low-electron energy region, the very high-$T_{\mathrm{loc}}$ can be found, which has a large influence on the temperature of the low-energy bulk region. Therefore, for a reference, $T_{e0}$ is defined as the average value of $T_{\mathrm{loc}}$ over the electron energy range 1≤ϵ[eV]≤2 for the discussion of overall behavior of electron temperature, which will be discussed later in Figure 14.

In addition, to compare various electron temperatures of the N${}_{2}$ plasma as the O${}_{2}$ plasma in the previous section, the electron entropy of the N${}_{2}$ plasma must be calculated. Figure 15 and Figure 16 show the electron entropy *S* and the electron mean energy *U* of the N${}_{2}$ plasma plotted against the reduced electric field, respectively. Thus, the entropy *S* is related with the electron internal energy *U* is shown in Figure 17, which gives $T_{e}^{\mathrm{st}}$ by Equation (Equation 2). Finally, the comparison of various electron temperatures of nitrogen plasma is shown in Figure 14.

Just like the oxygen plasma in the previous section, it is found that the statistical electron temperature $T_{e}^{\mathrm{st}}$ defined by the thermodynamic relation and the kinetic temperature $T_{e}^{\mathrm{kin}}$ showed intermediate values, which can be interpreted as the weighted average of the bulk electron temperature $T_{e0}-T_{e2}$ and the tail electron temperature $T_{\mathrm{loc}}\left(30\;\mathrm{eV}\right)$. In any case, it can be seen that these temperatures have different values. For the nitrogen plasma as well as for the oxygen plasma, it was found that
(39)$$T_{e}^{\mathrm{kin}}=\left(1.04-1.09\right)\times T_{e}^{\mathrm{st}}.$$
Similarly to the oxygen plasma case in Equation (Equation 38), $T_{e}^{\mathrm{kin}}$ is always larger than $T_{e}^{\mathrm{st}}$, and the discrepancy is found to be approximately (4–9)% within the examined parameter range of the reduced electric field.

In addition, it is confirmed that the relationship between *S* and lnU is also described approximately by Equation (Equation 20) for the nitrogen plasma, which is shown in Figure 18. This figure indicates that the entropy *S* increases almost linearly with increasing lnU, d(S/k)/d(lnU)≃ 1.53–1.58 ≩ 1.50. Although the excellent linear relationship can be confirmed between lnU and *S*, the proportional coefficient is found to be 1.53–1.58, which is always larger than the value 1.50, predicted by Equation (Equation 20). At present, it is still difficult to confirm the exact agreement, which will be further discussed in the next subsection.

### 3.3. Mathematical Discussion on the Results of the Relation between *S* and lnU
from Alvarez et al.’s Theory

In short, both for the oxygen and the nitrogen plasmas, the statistical electron temperature $T_{e}^{\mathrm{st}}$ approximately agrees but not exactly, with the kinetic electron temperature $T_{e}^{\mathrm{kin}}$. In addition, although the relationship between the electron entropy *S* and the logarithm of the electron energy lnU was found to be approximately linear, the coefficient did not agree with the one analyzed by Alvarez et al. [15]. Namely, the following equation was confirmed through this study:(40)$$S=S_{0}+C_{1}k\ln U,$$
where $C_{1}$ is the constant found for each discharge plasma, $C_{1}=1.55$–1.61 for oxygen plasma and $C_{1}=1.53$–1.58 for nitrogen plasma, while it had been predicted to be exactly $C_{1}=1.50$ for any discharge species based on the calculation by using BOLSIG+ software (Ver. 06/2013) [15].

As a small discussion on the implications of the obtained results, a comparison with a simplified model should be useful. In this paragraph, the electron mean energy U=〈ϵ〉 is assumed to increase by a factor exp(δ) in comparison with the original one due to the increase in the reduced electric field E/N, where the parameter δ is a constant with |δ|≪1. The original EEDF F(ϵ) is also assumed to satisfy Equations (Equation 15) and (Equation 16) for the mean energy and the entropy. The resultant change in the original EEDF F(ϵ) is assumed to be the following in a self-similar manner;
(41)$$F\left(\epsilon \right)\to F^{\prime }\left(\epsilon \right)=\exp\left(-\delta \right)F\left[\exp\left(-\delta \right)\epsilon \right].$$
At this transformation, the EEPF f(ϵ) should be carefully treated according to its definition as follows:(42)$$\begin{array}{c} f\left(\epsilon \right)\to f^{\prime }\left(\epsilon \right)=\frac{F^{\prime }\left(\epsilon \right)}{\sqrt{\epsilon }}=\exp\left(-\delta \right)\frac{F[\exp(-\delta )\epsilon ]}{\sqrt{\epsilon }}\;\\ \;\;\;\;\;\;\;=\exp\left(-\frac{3}{2}\delta \right)\frac{F[\exp(-\delta )\epsilon ]}{\sqrt{\exp(-\delta )\epsilon }}=\exp\left(-\frac{3}{2}\delta \right)f\left[\exp\left(-\delta \right)\epsilon \right]. \end{array}$$
Then, its normalization can be confirmed by the energy transform with $\epsilon^{\prime }\equiv \exp\left(-\delta \right)\epsilon$,
(43)$$\int_{0}^{\infty }F^{\prime }\left(\epsilon \right)d\epsilon =\int_{0}^{\infty }\exp\left(-\delta \right)F\left[\exp\left(-\delta \right)\epsilon \right]d\epsilon =\int_{0}^{\infty }F\left(\epsilon^{\prime }\right)d\epsilon^{\prime }=1,$$
this resultant electron mean energy $U^{\prime }$ is confirmed as
(44)$$\begin{array}{c} U^{\prime }=\int_{0}^{\infty }\epsilon F^{\prime }\left(\epsilon \right)d\epsilon =\int_{0}^{\infty }\epsilon \exp\left(-\delta \right)F\left[\exp\left(-\delta \right)\epsilon \right]d\epsilon \;\\ \;\;\;\;\;=\exp\left(\delta \right)\int_{0}^{\infty }\epsilon^{\prime }F\left(\epsilon^{\prime }\right)d\epsilon^{\prime }=\exp\left(\delta \right)U, \end{array}$$
which, indeed, supports the first assumption, i.e., the mean energy increase by a factor exp(δ). In the meantime, the resultant entropy $S^{\prime }$ is found as.
(45)$$\begin{array}{c} S^{\prime }=-k\int_{0}^{\infty }F^{\prime }\left(\epsilon \right)\ln\left[f^{\prime }\left(\epsilon \right)\right]d\epsilon \;\\ \;\;\;\;=-k\int_{0}^{\infty }\exp\left(-\delta \right)F\left[\exp\left(-\delta \right)\epsilon \right]\left\{-\frac{3}{2}\delta +\ln f\left[\exp\left(-\delta \right)\epsilon \right]\right\}d\epsilon \;\\ \;\;\;\;\;=-k\int_{0}^{\infty }F\left(\epsilon^{\prime }\right)\left[-\frac{3}{2}\delta +\ln f\left(\epsilon^{\prime }\right)\right]d\epsilon^{\prime }=S+\frac{3}{2}k\delta \end{array}$$
consequently, by this small variation in the EEDF with a self-similar condition, Equations (Equation 44) and (Equation 45) yield the following:(46)$$S^{\prime }=S+\frac{3}{2}k\ln\left(\frac{U^{\prime }}{U}\right).$$
Therefore, the statistical electron temperature ${T^{\prime }}_{e}^{\mathrm{st}}$ after the increase in the reduced electric field is calculated as
(47)$${T^{\prime }}_{e}^{\mathrm{st}}={\left(\frac{dS^{\prime }}{dU^{\prime }}\right)}^{-1}=\exp\left(\delta \right){\left(\frac{dS}{dU}\right)}^{-1}=\exp\left(\delta \right)T_{e}^{\mathrm{st}},$$
hence, the electron temperature also increases by a factor exp(δ), which is quantitatively consistent with Equation (Equation 44). Or, more analytically with Equations (Equation 44) and (Equation 45), the statistical temperature can directly obtained as follows:(48)$$\frac{1}{T_{e}^{\mathrm{st}}}=\frac{dS}{dU}=\lim_{\delta \to 0}\frac{S^{\prime }-S}{U^{\prime }-U}=\lim_{\delta \to 0}\frac{(3/2)\delta k}{U[\exp(\delta )-1]}=\frac{3k}{2U}=\frac{1}{T_{e}^{\mathrm{kin}}}.$$
Thus, the equation $U=\left(3/2\right)kT_{e}^{\mathrm{st}}$, or $T_{e}^{\mathrm{st}}=T_{e}^{\mathrm{kin}}$ is proved exactly for the EEDF variation formulated in Equation (Equation 41), whatever the EEDF is, for example any non-Maxwellian function. On the other hand, if some disagreement between them is detected, the EEDF considerably changes its dependence on the electron energy, at least into non-similar distribution. In the present study, minor difference between $T_{e}^{\mathrm{st}}$ and $T_{e}^{\mathrm{kin}}$, ∼2–9%, was detected. This indicates that the EEDF of these molecular discharge plasmas changes their energy dependence in the E/N-range surveyed in the present study.

Meanwhile, in the present study, the proportional constant $C_{1}$ in Equation (Equation 40) is found to be $C_{1}=$1.55–1.61 for the oxygen plasma and $C_{1}=1.53$–1.58 for the nitrogen plasma, both of which could be considered a little larger than the value $C_{1}=1.50$ predicted by Alvarez et al. [15], although the discrepancy is small indeed. Therefore, it is concluded that this discrepancy is attributed to the change in the EEDF as shown in Figure 1 and Figure 10.

It is considered that this discrepancy is originated from the difference in the governing equation to calculate the EEPF *f*. Of course, the EEPF, or the EEDF, is calculated from the Boltzmann equation, Equation (Equation 26). In this formulation, the Coulomb collision term is neglected owing to the low-ionization degree of the simulated plasmas, and consequently, the non-linear term due to the electron-electron collisions is ignored, although BOLSIG+ includes the Coulomb collisions as an option [14]. Unfortunately, the non-linear electron-electron collision processes cannot be treated in the present analysis. Concerning high-electron density limit, it is already found by many researchers that the EEDF becomes the Maxwellian, Equation (Equation 12) [3,9,28].

However, when the number densities of the target molecules are assumed to be constant and the chemical changing such as the oxygen dissociation or nitrogen vibrational excitation is neglected, this equation becomes linear with respect to the unknown function *f* by neglecting the Coulomb collision terms. That is, if the superelastic collision processes are ignored and the composition and density of the target molecules are fixed, Equation (Equation 31) becomes a homogeneous second-order linear ordinary differential equation as
(49)$$\frac{d^{2}f\left(\epsilon \right)}{d\epsilon^{2}}+a\left(\epsilon \right)\frac{df(\epsilon )}{d\epsilon }+b\left(\epsilon \right)f\left(\epsilon \right)=0,$$
where a(ϵ) and b(ϵ) are functions of electron energy ϵ. They are dependent on cross sections, number density of the target, and the reduced electric field, however, independent of the EEPF *f*, in traditional modelings [44]. Then, at least one of its solutions could be expressed as a power series, whereas another solution may require the Frobenius method to treat the regular singular point [45]. However for any situation, if Equation (Equation 49) should be adopted to calculate the EEPF, the Gibbs entropy, Equation (Equation 16), could have its corresponding physical meaning.

However, the real situation is much more complicated. If the variation of the target composition is introduced into the governing equations, i.e., if the dissociation of oxygen molecules with increasing reduced electric field is taken into account, the terms a(ϵ) and b(ϵ) should be dependent on the EEPF *f*, and what is more complicated, the dependence is naturally non-linear. This is also common for the nitrogen discharge where the superelastic collisions are introduced with the varying VDF of N${}_{2}$ X state according to the value of E/N. These conditions make the system of electron energy determination non-linear even when the two-term approximated Boltzmann equation is adopted to describe the EEPF. Consequently, the EEPF of such plasmas will not be an exponential function. Rather, it will be described by a power-law distribution [46]. For this type of distribution function, the corresponding entropy has been discussed well in the field of statistical and mathematical physics. For example, if the probability density function is given in terms of the *q*-exponential distribution [46,47] as
(50)$$F\left(x\right)=\left(2-q\right)\lambda e_{q}\left(-\lambda x\right),$$
where $e_{q}$ is the *q*-exponential [48]
(51)$$e_{q}\left(x\right)\equiv {\left[1+(1-q)x\right]}^{\frac{1}{1-q}},$$
the corresponding entropy is already found as the Tsallis entropy $S_{q}$ [48,49,50]:(52)$$S_{q}=-\frac{k}{1-q}\left[1-\int_{0}^{\infty }F{\left(x\right)}^{q}dx\right].$$
In short, the validity to calculate the entropy of the system should be reconsidered from the fundamental viewpoint for the applications to non-equilibrium plasmas kinetics, just like the discussion with Tsallis entropy for the analysis of energy distribution of cosmic rays [51]. That is, for the non-equilibrium system, it is concluded that the formulation of the entropy with the distribution function should be reconsidered from the viewpoint of statistical physics based on non-linear entropy, instead of the Gibbs entropy, Equation (Equation 16), which remains as an avenue future work.

## 4. Conclusions

The EEDF F(ϵ) of the weakly-ionized oxygen and nitrogen plasmas was calculated as a function of the reduced electric field E/N by solving the two-term approximated Boltzmann equation. In this study, the changes in the composition of neutral particles as collision counterparts of electrons were taken into account, and the Boltzmann equation was simultaneously solved with the changes in the chemical composition of the discharge media, so that it would be self-consistent as an electronic simulation of discharge plasma by solving the chemical kinetics of some essential heavy species in the discharge plasma. That is, in the oxygen plasma, the increase in the dissociation degree of the oxygen molecules was also considered with the increase in the reduced electric field. On the other hand, in the nitrogen plasma, the vibrational distribution function, VDF, N${}_{2}$ was solved simultaneously. In addition to the vibrational excitation as an electron inelastic collision, vibrational deexcitation as an electron superelastic collision was also incorporated into the Boltzmann equation.

Using the EEDF F(ϵ) obtained as described above, the electron mean energy $U=\langle \epsilon \rangle =\int_{0}^{\infty }\epsilon F\left(\epsilon \right)d\epsilon$ and the Gibbs entropy $S=-k\int_{0}^{\infty }F\left(\epsilon \right)\ln\left[f\left(\epsilon \right)\right]d\epsilon$ were calculated with the EEPF $f\left(\epsilon \right)\equiv F\left(\epsilon \right)/\sqrt{\epsilon }$. Then, the temperature determined in a statistical-thermodynamic way was calculated from the relationship between *U* and *S* as $T_{e}^{\mathrm{st}}={\left(\partial S/\partial U\right)}^{-1}$. Other possible candidates of the electron temperature were also calculated, such as the electron kinetic temperature $T_{e}^{\mathrm{kin}}$ defined as 2/(3k) times the average electron energy 〈ϵ〉, or the “local temperature” $T_{\mathrm{loc}}$ given as the slope of the EEPF f(ϵ) at a local electron energy ϵ.

For the oxygen plasma, two characteristic local electron temperatures were chosen to be $T_{e1}$ and $T_{e2}$, which were defined to be the maximum of $T_{\mathrm{loc}}$ and its mean value in the energy range ϵ≥8.4 eV, respectively. It was found that $T_{e1}>T_{e}^{\mathrm{kin}}>T_{e}^{\mathrm{st}}>T_{e2}$, which showed that $T_{e}^{\mathrm{st}}$ and $T_{e}^{\mathrm{kin}}$ are some kind of weighted average electron temperatures.

Meanwhile for the nitrogen plasma, as the local temperatures, $T_{e0},T_{e1}$ and $T_{e2}$ were defined to be the average $T_{\mathrm{loc}}$ for 1–2 eV, the minimum at approximately ϵ2.2 eV due to the N${}_{2}$ vibrational excitation and the maximum of the bulk electron energy region, respectively. It was found that $T_{e0}>T_{e2}>T_{e}^{\mathrm{kin}}>T_{e}^{\mathrm{st}}>T_{e1}>T_{\mathrm{loc}}\left(30\;\mathrm{eV}\right)$, which also indicated that $T_{e}^{\mathrm{st}}$ and $T_{e}^{\mathrm{kin}}$ are weighted average electron temperatures.

For both plasmas, although the good linear relationship between the entropy *S* and the logarithm of the electron energy lnU was confirmed, it was found that $d{\left(S/k\right)}_{d}\left(\ln U\right)≃$ 1.55–1.61 for the oxygen plasma and ∼1.53–1.58 for the nitrogen plasma, which agreed approximately with the predicted value by Alvarez et al., as 1.50, but not exactly. It was also found that the statistical-thermodynamic electron temperature $T_{e}^{\mathrm{st}}$, which is calculated from the relation between the entropy and the mean energy, approximately agrees with the electron kinetic temperature $T_{e}^{\mathrm{kin}}$, but not exactly. This was different from the result of Alvarez et al., who concluded $T_{e}^{\mathrm{st}}=T_{e}^{\mathrm{kin}}$, which was also proved in the present at least for the case where the EEDF changes in a self-similar manner. On the other hand, the non-self-similar variation in the EEDF could cause the discrepancy from the results found by Alvarez et al., where they did not consider the variation in the chemical composition of the target molecules. Namely, they considered neither the dissociation of oxygen molecules into oxygen atoms, nor the energy exchange with the vibrational levels of nitrogen molecules by inelastic and superelastic collisions. The major difference in this study lies in the fact that the dissociation or the chemical variation of the target molecules are included in the calculation of the EEDF. That is, when solving the Boltzmann equation to obtain EEDF, the influence of EEDF should be appropriately incorporated into the collision term of the Boltzmann equation and the change in electron energy should be examined. It was discussed that by doing so, the two-term approximated Boltzmann equation, which was linear with respect to the unknown function EEPF by ignoring electron-electron collisions, f(ϵ), becomes non-linear. Consequently, it was also confirmed that the corresponding entropy is not necessarily the Gibbs entropy treated in this study, other type of non-equilibrated entropy should be also discussed, which should be further examined in the future.

## Figures and Tables

**Figure 1 entropy-25-00276-f001:**
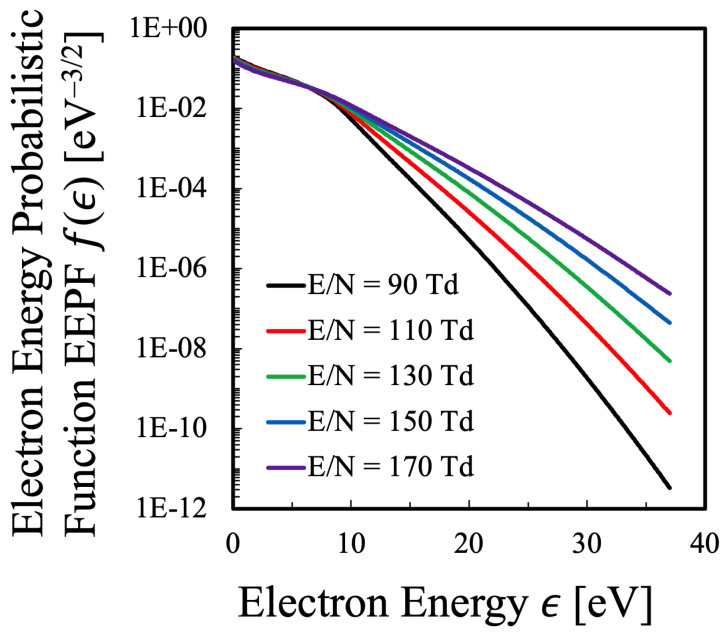
The electron energy probabilistic function (EEPF) of the oxygen plasma obtained by the method described in Section 2.3 with respect to reduced electric field E/N.

**Figure 2 entropy-25-00276-f002:**
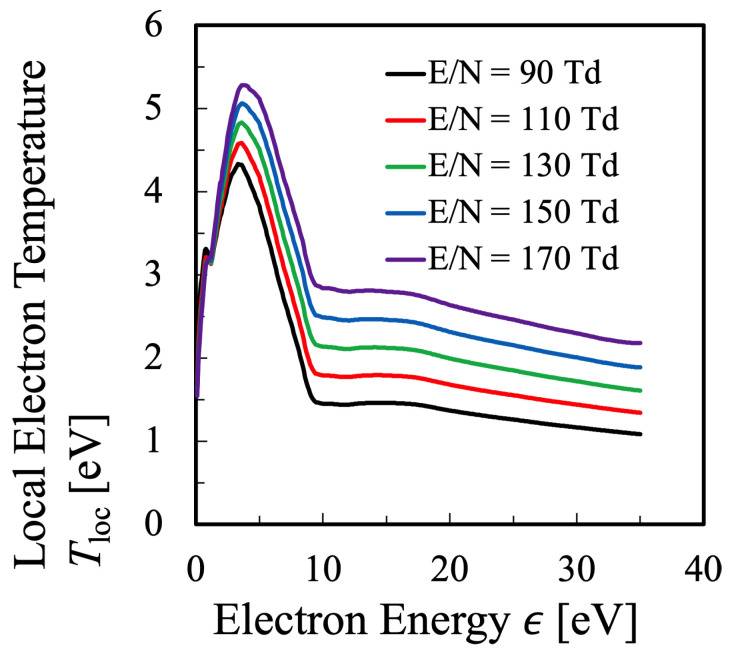
Dependence of the local electron temperature in energy space $T_{\mathrm{loc}}$ of the oxygen plasma as determined from Equation (Equation 36) with the data shown in Figure 1 for several values of the reduced electric field.

**Figure 3 entropy-25-00276-f003:**
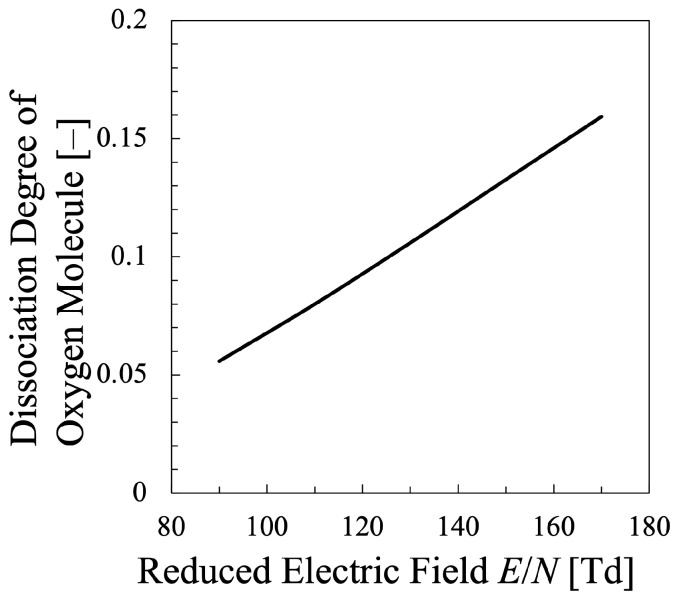
Dissociation degree of oxygen molecules in the oxygen plasma plotted against the reduced electric field for constant electron density $N_{e}=2\times 10^{11}\;{\mathrm{cm}}^{-3}$.

**Figure 4 entropy-25-00276-f004:**
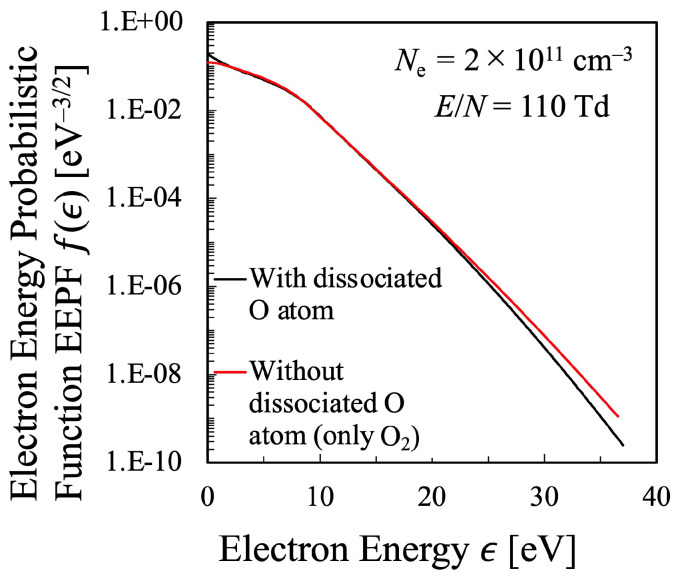
Comparison of the calculated electron energy probabilistic function (EEPF) f(ϵ) of the oxygen plasma with and without the existence of the dissociated oxygen atoms in the Boltzmann Equation (Equation 31), The number density of oxygen atoms is simultaneously solved by the method described in Section 2.3.

**Figure 5 entropy-25-00276-f005:**
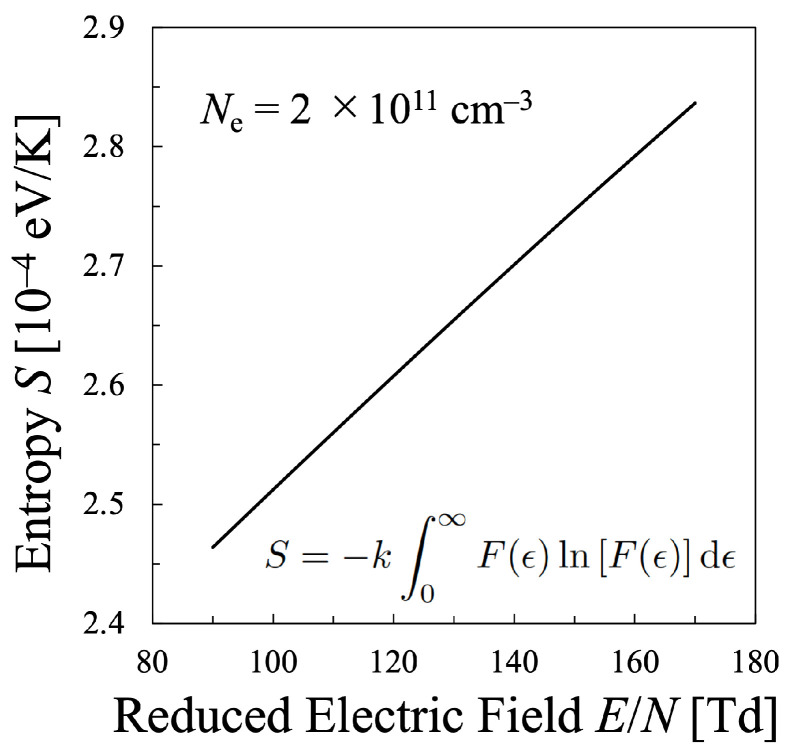
Entropy of the electrons in the oxygen plasma calculated by Equation (Equation 16) for constant electron density $N_{e}=2\times 10^{11}\;{\mathrm{cm}}^{-3}$.

**Figure 6 entropy-25-00276-f006:**
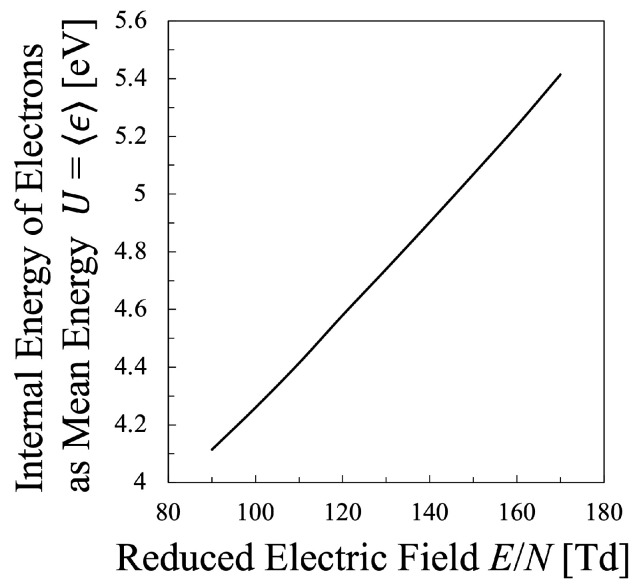
Internal energy of the electrons in the oxygen plasma as the electron mean energy U=〈ϵ〉, calculated by Equation (Equation 15) for constant electron density $N_{e}=2\times 10^{11}\;{\mathrm{cm}}^{-3}$.

**Figure 7 entropy-25-00276-f007:**
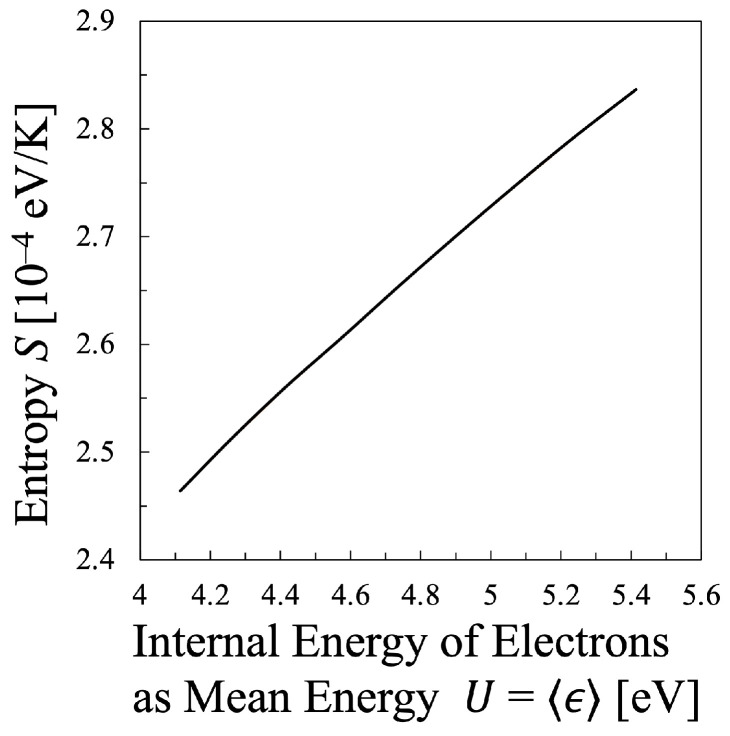
Entropy of the electrons *S* plotted against the internal energy of the electrons in the oxygen plasma for constant electron density $N_{e}=2\times 10^{11}\;{\mathrm{cm}}^{-3}$.

**Figure 8 entropy-25-00276-f008:**
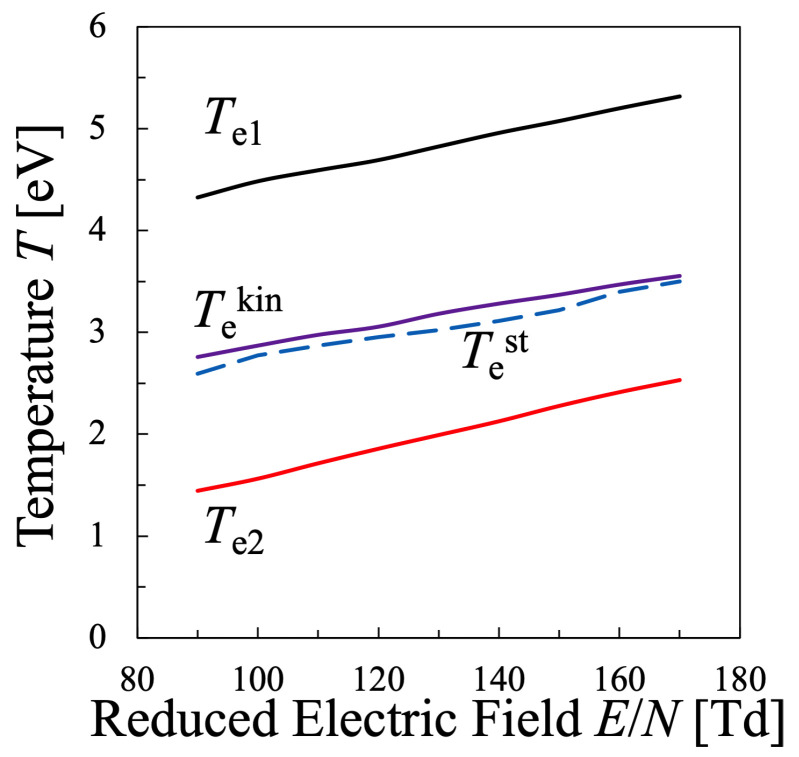
Comparison between various electron temperatures of the oxygen plasma defined in the present study. The electron density is set as $N_{e}=2\times 10^{11}\;{\mathrm{cm}}^{-3}$.

**Figure 9 entropy-25-00276-f009:**
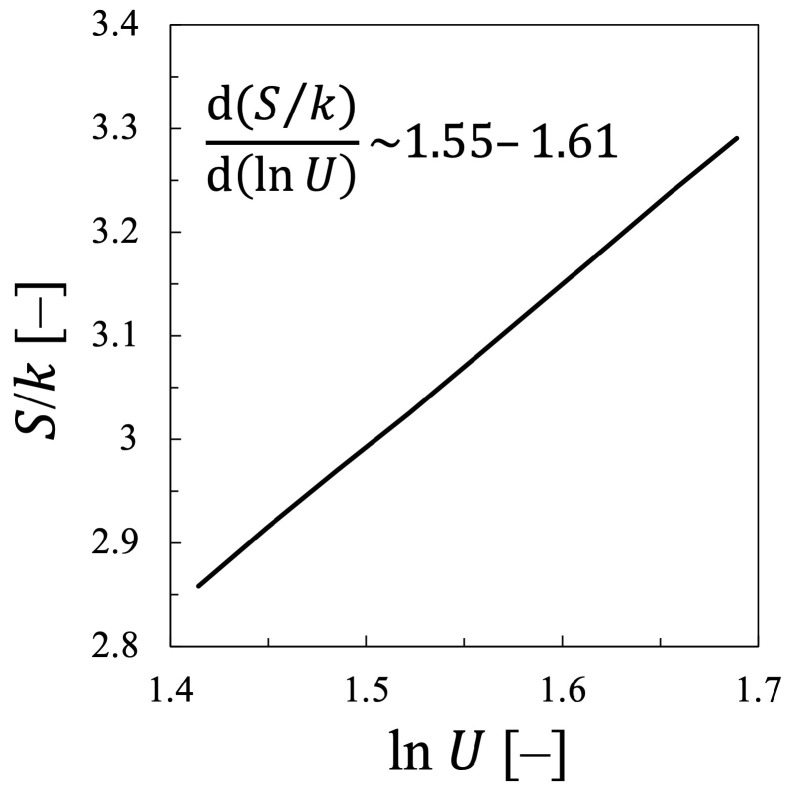
Entropy of the electrons *S* plotted against the logarithms of the internal energy of the electrons in the oxygen plasma lnU, i.e., the same one as Figure 7 but with the logarithmic scale of the horizontal axis.

**Figure 10 entropy-25-00276-f010:**
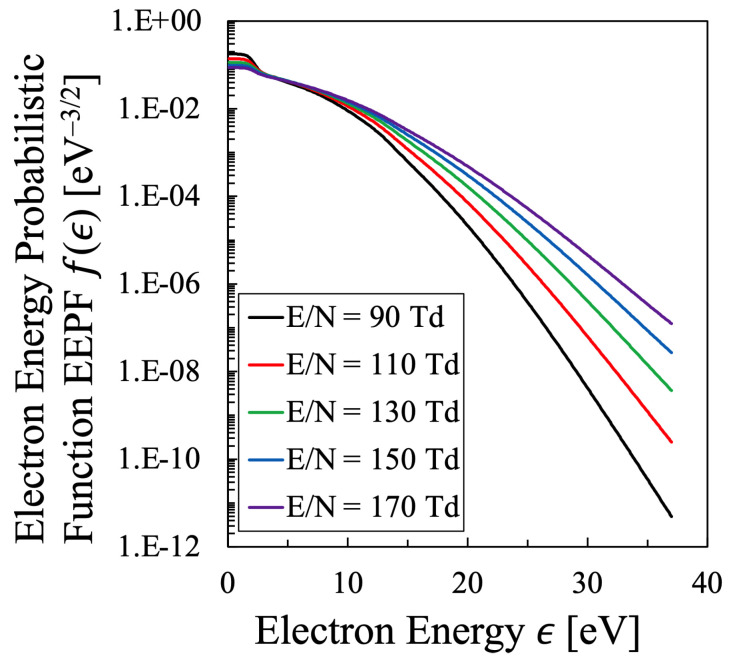
The electron energy probabilistic function (EEPF) of the nitrogen plasma obtained by the method described in Section 2.4 with respect to reduced electric field E/N. The electron density is set to be constant, $N_{e}=4.0\times 10^{11}\;{\mathrm{cm}}^{-3}$.

**Figure 11 entropy-25-00276-f011:**
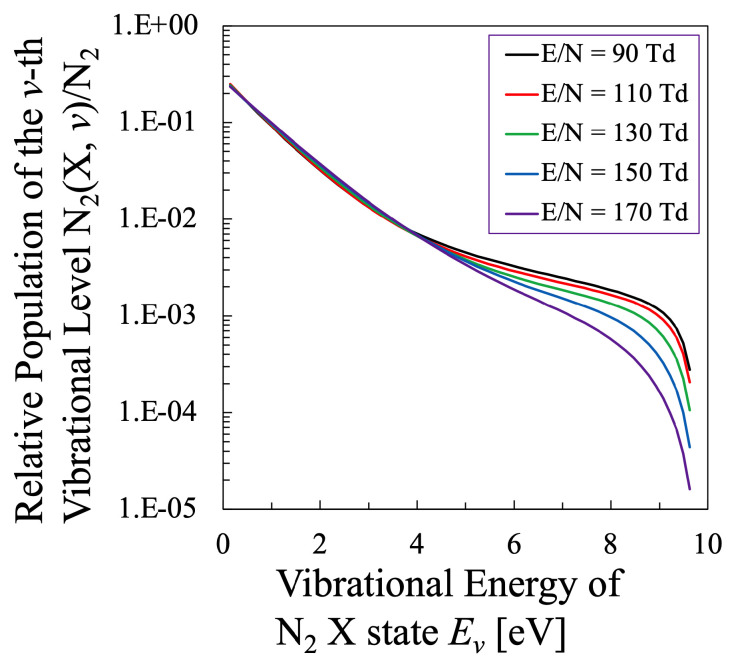
The relative population density of the *v*-th Vibrational Level of N${}_{2}$ (X, *v*) state in the N${}_{2}$ (X) state plotted against the reduced electric field E/N.

**Figure 12 entropy-25-00276-f012:**
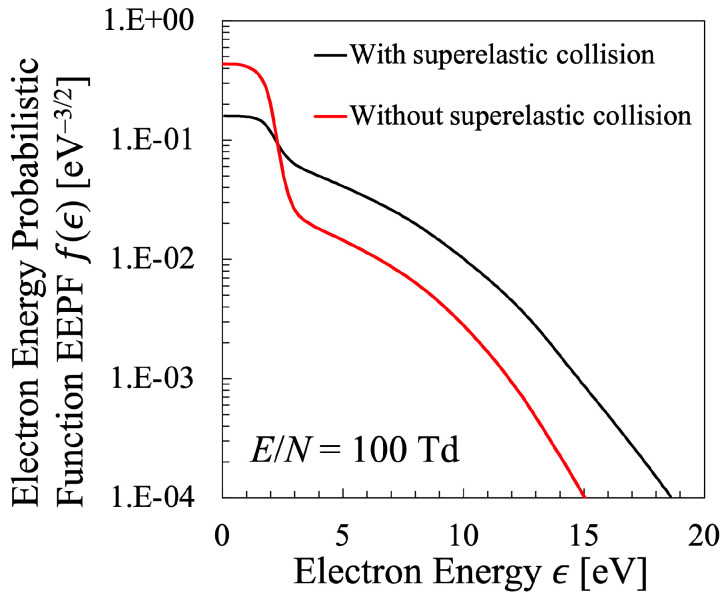
Comparison between the EEPF with and without superelastic collision for nitrogen plasma with the reduced electric field E/N=100 Td [26].

**Figure 13 entropy-25-00276-f013:**
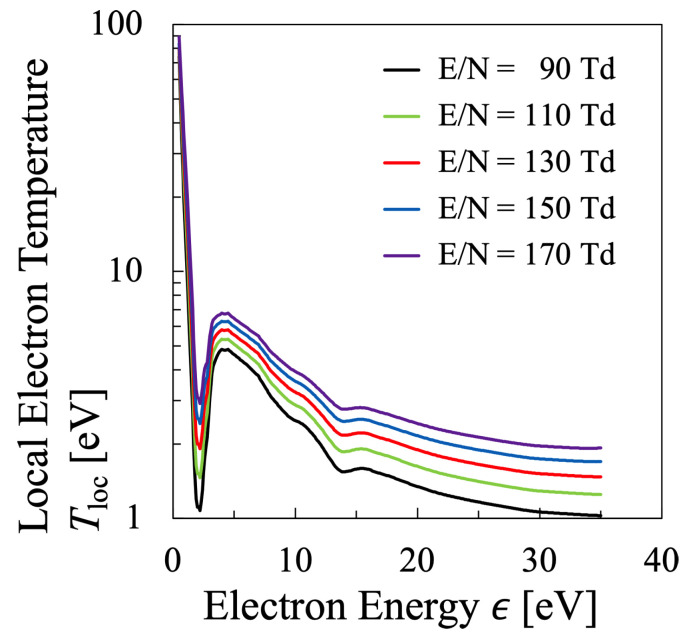
Dependence of the local electron temperature in energy space $T_{\mathrm{loc}}$ of the nitrogen plasma as determined from Equation (Equation 36) with the data shown in Figure 10 for several values of the reduced electric field.

**Figure 14 entropy-25-00276-f014:**
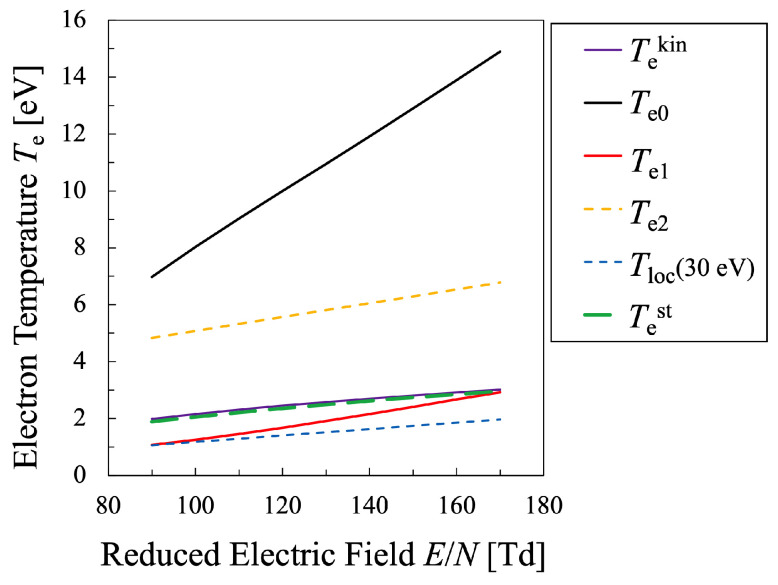
Comparison between various electron temperatures of the nitrogen plasma defined in the present study. The electron density is set as $N_{e}=4\times 10^{11}\;{\mathrm{cm}}^{-3}$.

**Figure 15 entropy-25-00276-f015:**
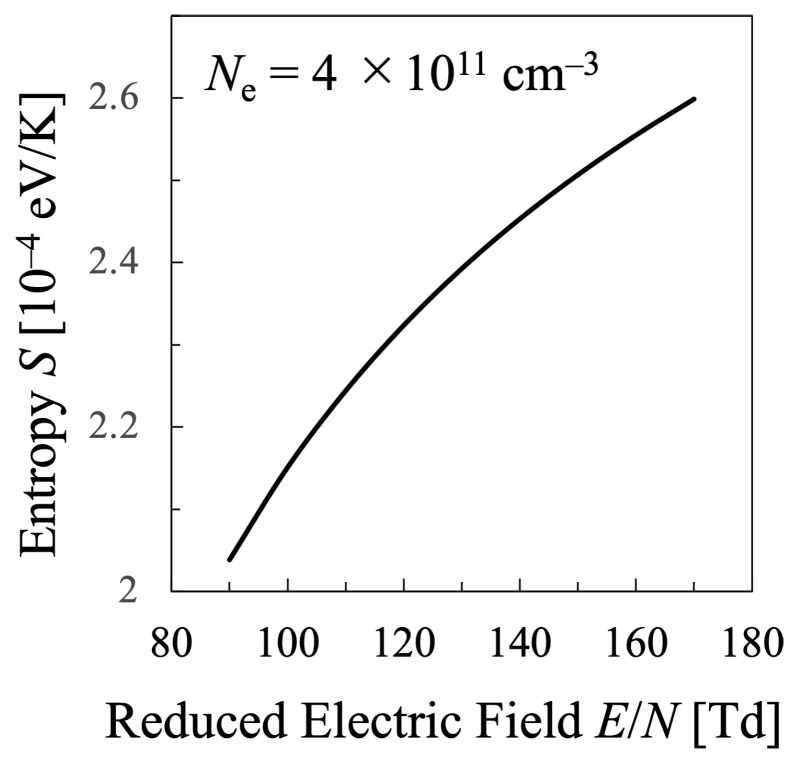
Entropy of the electrons in the nitrogen plasma calculated by Equation (Equation 16) for constant electron density $N_{e}=4\times 10^{11}\;{\mathrm{cm}}^{-3}$.

**Figure 16 entropy-25-00276-f016:**
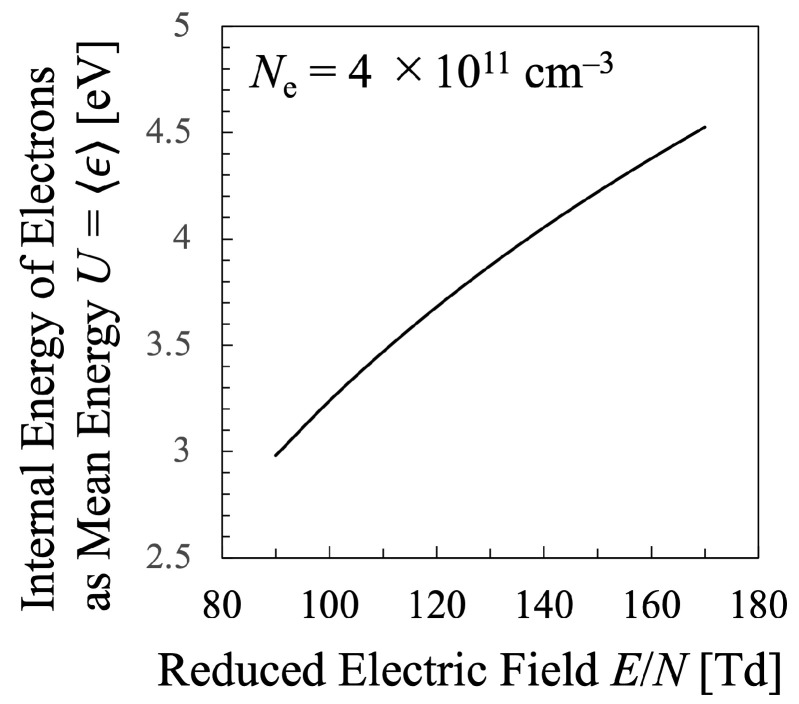
Internal energy of the electrons in the nitrogen plasma as the electron mean energy U=〈ϵ〉, calculated by Equation (Equation 15) for constant electron density $N_{e}=4\times 10^{11}\;{\mathrm{cm}}^{-3}$.

**Figure 17 entropy-25-00276-f017:**
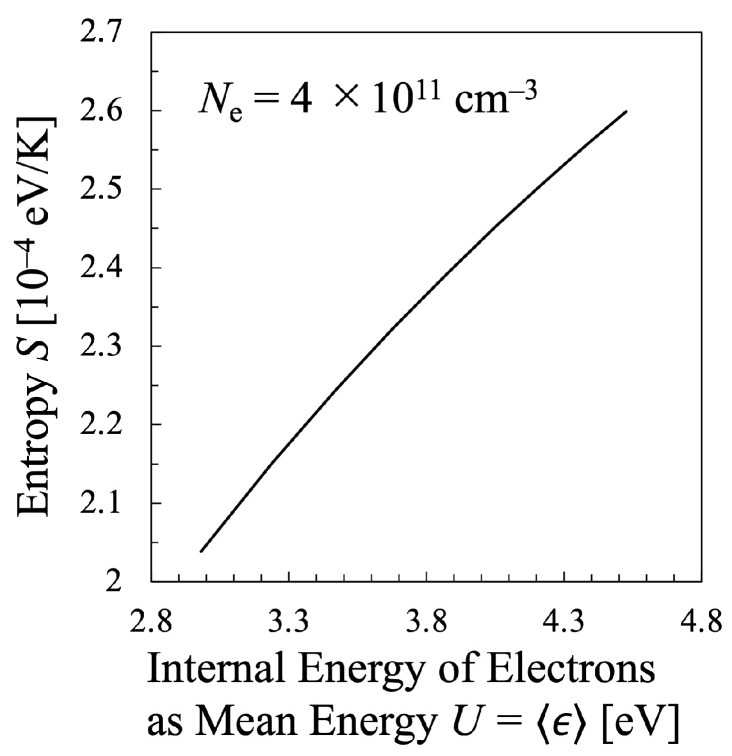
Entropy of the electrons *S* plotted against the internal energy of the electrons in the nitrogen plasma for constant electron density $N_{e}=4\times 10^{11}\;{\mathrm{cm}}^{-3}$.

**Figure 18 entropy-25-00276-f018:**
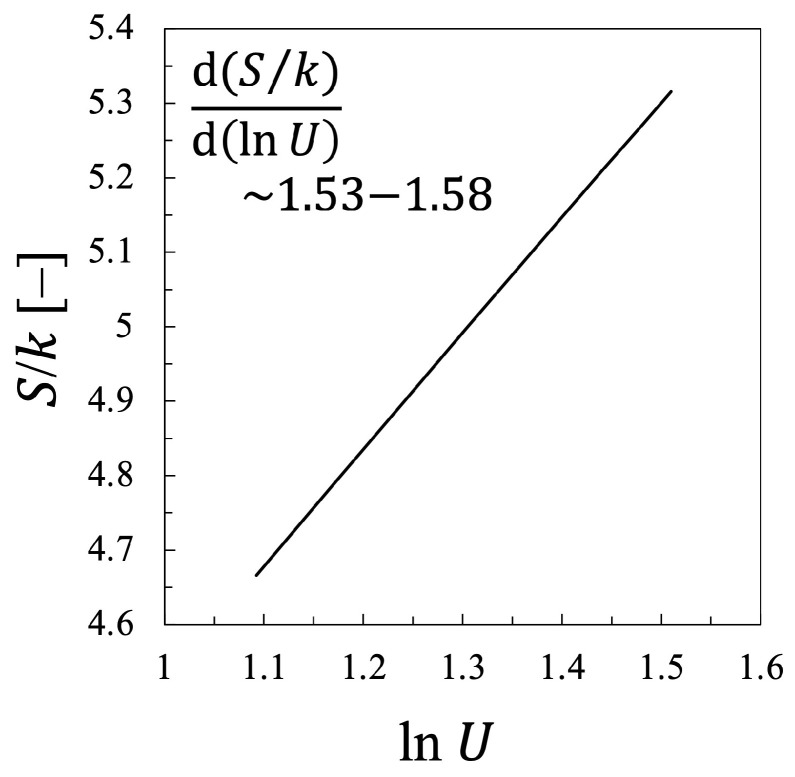
Entropy of the electrons *S* plotted against the logarithms of the internal energy of the electrons in the nitrogen plasma lnU, i.e., the same one as Figure 17 but with the logarithmic scale of the horizontal axis.

**Table 1 entropy-25-00276-t001:** List of electron inelastic collision processes considered in the calculation of number densities of excited states in Equation (Equation 27).

Number	Electron Collision Reactions		References
1	$O_{2}\;\left(X{\;}^{3}\Sigma_{g}^{-}\right)$ $\;+\;e^{-}$	⇄ $O_{2}\;\left(a{\;}^{1}\Delta_{g}\right)$ $+\;e^{-}$	[7]
2	$O_{2}\;\left(X{\;}^{3}\Sigma_{g}^{-}\right)$ $\;+\;e^{-}$	⇄ $O_{2}\;\left(b{\;}^{1}\Sigma_{g}^{+}\right)$ $+\;e^{-}$	[7]
3	$O_{2}\;\left(a{\;}^{1}\Delta_{g}\right)$ $\;+\;e^{-}$	⇄ $O_{2}\;\left(b{\;}^{1}\Sigma_{g}^{+}\right)$ $+\;e^{-}$	[20]
4	$O_{2}\;\left(X{\;}^{3}\Sigma_{g}^{-}\right)$ $\;+\;e^{-}$	→ $O_{2}^{+}$ $+\;2\;e^{-}$	[7]
5	$O_{2}\;\left(X{\;}^{3}\Sigma_{g}^{-}\right)$ $\;+\;e^{-}$	⇄ $O^{-}$ $+\;O\;{(}^{3}P)$	[40]
6	$O_{2}\;\left(a{\;}^{1}\Delta_{g}\right)$ $\;+\;e^{-}$	⇄ $O^{-}$ $+\;O\;{(}^{3}P)$	[7,40]
7	$O_{2}\;\left(X{\;}^{3}\Sigma_{g}^{-}\right)$ $\;+\;e^{-}$	⇄ $2\;O\;{(}^{3}P)$ $+\;e^{-}$	[20]
8	$O_{2}\;\left(a{\;}^{1}\Delta_{g}^{-}\right)$ $\;+\;e^{-}$	⇄ $2\;O\;{(}^{3}P)$ $+\;e^{-}$	[7,20]
9	$O_{2}^{+}$ $\;+\;e^{-}$	→ $2\;O\;{(}^{3}P)$	[7]
10	$O_{3}$ $\;+\;e^{-}$	⇄ $O\;{(}^{3}P)+O_{2}\;\left(X{\;}^{3}\Sigma_{g}^{-}\right)$ $+\;e^{-}$	[19]
11	$O^{-}$ $\;+\;e^{-}$	→ $O\;{(}^{3}P)$ $+\;2\;e^{-}$	[3]
12	$O_{2}\;\left(X{\;}^{3}\Sigma_{g}^{-}\right)$ $\;+\;e^{-}$	→ $O\;{(}^{3}P)+O\;{(}^{1}D)$ $+\;e^{-}$	[40]
13	$O_{2}\;\left(a{\;}^{1}\Delta_{g}\right)$ $\;+\;e^{-}$	→ $O\;{(}^{3}P)+O\;{(}^{1}D)$ $+\;e^{-}$	[7,40]
14	$O\;{(}^{3}P)$ $\;+\;e^{-}$	⇄ $O\;{(}^{1}D)$ $+\;e^{-}$	[7]
15	$O_{2}\;\left(X{\;}^{3}\Sigma_{g}^{-}\right)$ $\;+\;e^{-}$	→ $2\;O\;{(}^{1}D)$ $+\;e^{-}$	[40]

**Table 2 entropy-25-00276-t002:** List of atomic and molecular collision processes considered in the calculation of number densities of excited states in Equation (Equation 27).

Number	Atomic or Molecular Collision Reactions		References
16	$O_{2}\;\left(a{\;}^{1}\Delta_{g}\right)$ $\;+\;O^{-}$	→ $O_{3}+\;e^{-}$	[7]
17	$O_{2}\;\left(b{\;}^{1}\Sigma_{g}^{+}\right)$ $\;+\;O^{-}$	→ $O\;{(}^{3}P)+O_{2}\;\left(X{\;}^{3}\Sigma_{g}^{-}\right)+\;e^{-}$	[7]
18	$O^{-}$ $\;+\;O_{2}^{+}$	→ $O\;{(}^{3}P)+O_{2}\;\left(X{\;}^{3}\Sigma_{g}^{-}\right)$	[7]
19	$O_{2}\;\left(a{\;}^{1}\Delta_{g}\right)$ $\;+\;O\;{(}^{3}P)$	⇄ $O_{2}\;\left(X{\;}^{3}\Sigma_{g}^{-}\right)+\;O\;{(}^{3}P)$	[7]
20	$O_{3}$ $\;+\;O_{2}\;\left(X{\;}^{3}\Sigma_{g}^{-}\right)$	⇄ $2\;O_{2}\;\left(X{\;}^{3}\Sigma_{g}^{-}\right)+\;O\;{(}^{3}P)$	[7]
21	$O\;{(}^{3}P)$ $\;+\;O_{3}$	⇄ $2\;O\;{(}^{3}P)+\;O_{2}\;\left(X{\;}^{3}\Sigma_{g}^{-}\right)$	[7]
22	$O\;{(}^{3}P)$ $\;+\;O_{3}$	→ $O_{2}\;\left(a{\;}^{1}\Delta_{g}\right)+O_{2}\;\left(X{\;}^{3}\Sigma_{g}^{-}\right)$	[7]
23	$O_{2}\;\left(b{\;}^{1}\Sigma_{g}^{+}\right)$ $\;+\;O_{3}$	→ $2\;O_{2}\;\left(X{\;}^{3}\Sigma_{g}^{-}\right)+O\;{(}^{3}P)$	[19]
24	$O_{2}\;\left(a{\;}^{1}\Delta_{g}\right)$ $\;+\;O_{2}\;\left(X{\;}^{3}\Sigma_{g}^{-}\right)$	⇄ $2\;O_{2}\;\left(X{\;}^{3}\Sigma_{g}^{-}\right)$	[7]
25	$O_{2}\;\left(a{\;}^{1}\Delta_{g}\right)$ $\;+\;O_{3}$	→ $2\;O_{2}\;\left(X{\;}^{3}\Sigma_{g}^{-}\right)+O\;{(}^{3}P)$	[19]
26	$O\;{(}^{3}P)$ $\;+\;O_{3}$	→ $2\;O_{2}\;\left(X{\;}^{3}\Sigma_{g}^{-}\right)$	[7]
27	$O^{-}$ $\;+\;O_{2}^{+}$	→ $3\;O\;{(}^{3}P)$	[7]
28	$O\;{(}^{1}D)$ $\;+\;O\;{(}^{3}P)$	→ $2\;O\;{(}^{3}P)$	[3]
29	$O\;{(}^{1}D)$ $\;+\;O_{2}\;\left(X{\;}^{3}\Sigma_{g}^{-}\right)$	→ $O\;{(}^{3}P)+O_{2}\;\left(X{\;}^{3}\Sigma_{g}^{-}\right)$	[3]
30	$O\;{(}^{1}D)$ $\;+\;O_{2}\;\left(X{\;}^{3}\Sigma_{g}^{-}\right)$	→ $O\;{(}^{3}P)+O_{2}\;\left(a{\;}^{1}\Delta_{g}\right)$	[3]

**Table 3 entropy-25-00276-t003:** List of inelastic collision processes of electrons considered in the calculation of the Boltzmann equation, Equation (Equation 31), for the oxygen plasma.

Number	Electron Inelastic Collision Reactions			Reference
31	$O_{2}\;\left(X{\;}^{3}\Sigma_{g}^{-}\right)+e^{-}$	→	$O_{2}\;\left(Y\right)+e^{-}$	[40]
	$Y=a{\;}^{1}\Delta_{g},b{\;}^{1}\Sigma_{g}^{+},4.5\;\mathrm{eV},6.0\;\mathrm{eV},8.4\;\mathrm{eV},9.97\;\mathrm{eV},14.7\;\mathrm{eV}$			
32	$O\;{(}^{3}P)+e^{-}$	→	$O\;\left(Z\right)+e^{-}$	[7]
	$Z={\;}^{1}D,{\;}^{1}S,3s{\;}^{5}S^{o},3s{\;}^{3}S^{o},3p{\;}^{5}P,3p{\;}^{3}P$			
33	$O_{2}\;\left(X{\;}^{3}\Sigma_{g}^{-}\right)+e^{-}$	→	$O_{2}^{+}+e^{-}+e^{-}$	[7]
34	$O\;{(}^{3}P)+e^{-}$	→	$O^{+}+e^{-}+e^{-}$	[41]
35	$O_{2}\;\left(X{\;}^{3}\Sigma_{g}^{-}\right)+e^{-}$	→	$O+O+e^{-}$	[40]
36	$O_{2}\;\left(X{\;}^{3}\Sigma_{g}^{-};v=0\right)+e^{-}$	→	$O_{2}\;\left(X{\;}^{3}\Sigma_{g}^{-};v=1,2\right)+e^{-}$	[40]

**Table 4 entropy-25-00276-t004:** List of collision processes relevant to the vibrational kinetics described in Equation (Equation 32).

Reaction	Vibrational Collision Reactions of N${}_{2}$		References
e-V	$N_{2}\;\left(X{\;}^{1}\Sigma_{g}^{+};v=0-8\right)+e^{-}$	⇄ $N_{2}\;\left(X{\;}^{1}\Sigma_{g}^{+};w=0-8\neq v\right)+e^{-}$	[42]
V-V	$N_{2}\;\left(X{\;}^{1}\Sigma_{g}^{+};v\right)+N_{2}\;\left(X{\;}^{1}\Sigma_{g}^{+};w\right)$	⇄ $N_{2}\;\left(X{\;}^{1}\Sigma_{g}^{+};v+1\right)+N_{2}\;\left(X{\;}^{1}\Sigma_{g}^{+};w-1\right)$	[7]
V-T	$N_{2}\;\left(X{\;}^{1}\Sigma_{g}^{+};v\right)+N_{2}$	⇄ $N_{2}\;\left(X{\;}^{1}\Sigma_{g}^{+};v-1\right)+N_{2}$	[7]
V-Diss.	$N_{2}\;\left(X{\;}^{1}\Sigma_{g}^{+};v\right)+N_{2}\;\left(X{\;}^{1}\Sigma_{g}^{+};w\right)$	→ $2N\left(2p{\;}^{4}S^{o}\right)+N_{2}\;\left(X{\;}^{1}\Sigma_{g}^{+};w-1\right)$	[7]

**Table 5 entropy-25-00276-t005:** List of inelastic collision processes of electrons considered in the calculation of the Boltzmann equation, Equation (Equation 31), for the nitrogen plasma.

Number	Electron Inelastic Collision Reactions		References
(1)	$N_{2}\;\left(X{\;}^{1}\Sigma_{g}^{+}\right)+e^{-}$	→ $N_{2}\;\left(Y\right)+e^{-}$	[7]
	$Y=A{\;}^{3}\Sigma_{u}^{+},B{\;}^{3}\Pi_{g},C{\;}^{3}\Pi_{u},a^{\prime }{\;}^{1}\Sigma_{u}^{-},a{\;}^{1}\Pi_{g},w{\;}^{1}\Delta_{u},B^{\prime }{\;}^{3}\Sigma_{u}^{-},W{\;}^{3}\Delta_{u}$		
(2)	$N_{2}\;\left(X{\;}^{1}\Sigma_{g}^{+}\right)+e^{-}$	→ $N_{2}^{+}\;\left(X{\;}^{2}\Sigma_{g}^{+}\right)+2e^{-}$	[7]
(3)	$N_{2}\;\left(X{\;}^{1}\Sigma_{g}^{+};v\right)+e^{-}$	⇄ $N_{2}\;\left(X{\;}^{1}\Sigma_{g}^{+};w\left[\neq v\right]\right)+e^{-}$	[42]
	(v,w=0−8)		
(4)	$N_{2}\;\left(X{\;}^{1}\Sigma_{g}^{+};v\right)+e^{-}$	→ $2N\;\left(2p{\;}^{4}S^{o}\right)+e^{-}$	[43]

## Data Availability

The data that support the findings of this study are available from the corresponding author upon reasonable request.

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
