# Peer review of "Discussion on Electron Temperature of Gas-Discharge Plasma with Non-Maxwellian Electron Energy Distribution Function Based on Entropy and Statistical Physics"

_entropy, 2023, doi:10.3390/e25020276_

Round 1

Reviewer 2 Report

Dear authors, editor,

I reviewed the article “Discussion …” H. Akatsuka and Y. Tanaka, for consideration of publication in Entropy/MDPI.  There are several concerns regarding clarity and contents. Some of specific concerns a re listed below, in random order.

Due to several ambiguities, it is challenging to assess the intent of the authors regarding the first words in the abstract “Electron temperature is reconsidered …” First of all, there needs to be a reasonable definition of temperature and/or the choice of state variables. Legendre transformations are usually employed for thermodynamic descriptions, and it should be made clear what is the thermodynamic potential that the authors utilize. For example, if arguments are made regarding temperature (line 122) then perhaps Free Energy is the preferred thermodynamic potential. Or is it the Gibbs potential, as you talk about pressure and temperature (line 123) Moreover, it will be required to actually define the meaning of chemical potential.

Abstract, line 2: please use SI units, and state the range of electron density.

Line 24: Please be specific: What is “low” in low-temperature and low-density? Perhaps include some numbers.

Line 28: “electrons … do …”

Line 30: Please be specific about “general experimental laboratories.” Possibly include a reference or one example

Line 140: “Boltzmann equation” is mentioned  seven times ( 2 times in abstract) up to line 140: Please state the integro-differential Boltzmann equation and discuss the specific meaning of the terms, including approximations that you assume – here the treatment of the collision term is important.

Line 142: What is “Boltzmann analysis” ?

Line 146: Or just below line 146: Please expand on “they solved” – there is need to discuss the actual integro-differential Boltzmann equation! Who is “they” in the sentence just above Equation (9)?

Line 205: There is reference to collision, how does this relate to the full Boltzmann equation?

Line 208: Or just above line 208: Equation (16) needs to be discussed further: Where does this come from? You call equation (16) “Boltzmann equation” just below 439.

With all the mentioned ambiguities, I strongly suggest major clarification and rewrite. Without a clearly presented mathematical and computational framework, it is next to impossible for readers to ascertain the value of the computational results presented in pages following page 4.

Very Respectfully,

Round 2

Reviewer 1 Report

The authors made sufficient revisions corresponding to my previous comments. I think this article is now in a satisfactory level for publication in Entropy.

Reviewer 2 Report

I appreciate the additions related to the integro-differential Boltzmann equation, and a few other edits in response to my comments.